# LncRNA GAS5 inhibits microglial M2 polarization and exacerbates demyelination

Dingya Sun[1,†] iD, Zhongwang Yu[1,†], Xue Fang[1,†], Mingdong Liu[1], Yingyan Pu[1], Qi Shao[1], Dan Wang[1], Xiaolin Zhao[1], Aijun Huang[1], Zhenghua Xiang[1], Chao Zhao[2], Robin JM Franklin[2], Li Cao[1,*] iD & Cheng He[1,**] iD

## Abstract

The regulation of inflammation is pivotal for preventing the development or reoccurrence of multiple sclerosis (MS). A biased ratio of high-M1 versus low-M2 polarized microglia is a major pathological feature of MS. Here, using microarray screening, we identify the long noncoding RNA (lncRNA) GAS5 as an epigenetic regulator of microglial polarization. Gain- and loss-of-function studies reveal that GAS5 suppresses microglial M2 polarization. Interference with GAS5 in transplanted microglia attenuates the progression of experimental autoimmune encephalomyelitis (EAE) and promotes remyelination in a lysolecithin-induced demyelination model. In agreement, higher levels of GAS5 are found in amoeboid-shaped microglia in MS patients. Further, functional studies demonstrate that GAS5 suppresses transcription of TRF4, a key factor controlling M2 macrophage polarization, by recruiting the polycomb repressive complex 2 (PRC2), thereby inhibiting M2 polarization. Thus, GAS5 may be a promising target for the treatment of demyelinating diseases.

**Keywords** demyelination; GAS5; M1/M2 polarization; microglia; multiple sclerosis
**Subject Categories** Immunology; Molecular Biology of Disease; Transcription

## Introduction

Multiple sclerosis (MS) is a multifocal, inflammatory demyelinating disease of the central nervous system (CNS). In the early stages, clinical signs appear in a relapsing–remitting pattern, but in the later stage, the disease develops into a secondary progressive phase without remission [1]. Currently, no effective treatment is available for progressive MS, and new therapeutic strategies are greatly needed [2,3].

Multiple sclerosis is an autoimmune disease with a significant neurodegenerative component [4,5]. As the major innate immune cells in the CNS, microglia play a central role in the inflammatory process of MS and in the disruption of neuronal elements within the CNS. Activated microglia can be classified into two functional subtypes: M1/classic and M2/alternative polarization [6]. Although such a classification underestimates the complexity of macrophage/microglia plasticity, the distinction nevertheless provides a useful framework for exploring the diverse functions of the innate immune system in disease pathogenesis. M1 polarized microglia promote neuronal apoptosis and inhibit oligodendrocyte progenitor cell (OPC) differentiation into mature oligodendrocytes (OLs), whereas M2 polarized microglia promote neuronal survival, neurite outgrowth, and OPC differentiation [7–10]. In experimental autoimmune encephalomyelitis (EAE), animal model of MS, inhibiting the secretion of inflammatory factors from microglia significantly, reduces the EAE severity, whereas transplanting M2 polarized microglia into the CNS promotes recovery [11–13]. In MS patients, M1 microglia were present in a variety of lesions, whereas M2 microglia appear only in acute lesions and the active edges of chronic active lesions where remyelination often occurs [9]. A delay in the transition to an M2 dominant environment is associated with the slowing of remyelination in aging [9,14]. Thus, microglial polarization is closely related to MS pathogenesis.

Long noncoding RNAs (lncRNAs) are a class of RNAs greater than 200 bp in length with no protein-coding capacity. Increasing evidence has shown that lncRNAs are involved in multiple biological processes including CNS disease [15,16]. However, no lncRNAs have been reported to play a role in the regulation of microglia and MS pathogenesis.

In the present study, more than one hundred lncRNAs were identified to be differentially expressed in M2 polarized microglia versus resting microglia through microarray screening. Among these lncRNAs, was the lncRNA growth arrest-specific 5 (GAS5), originally identified in growth-arrested cells and which encodes no functional protein but the introns encode a variety of functional small noncoding nucleolar RNAs (snoRNAs) [17]. GAS5 inhibits the proliferation of T cells and promotes tumor cell apoptosis [18–20]. In this study, we found that GAS5 suppressed IRF4 transcription by

---

1 Institute of Neuroscience, Key Laboratory of Molecular Neurobiology of the Ministry of Education and the Collaborative Innovation Center for Brain Science, Second Military Medical University, Shanghai, China
2 Wellcome Trust-Medical Research Council Stem Cell Institute, University of Cambridge, Cambridge, UK
*Corresponding author. Tel: +86 21 81871042 516; Fax: +86 21 65492132; E-mail: caoli@smmu.edu.cn
**Corresponding author. Tel: +86 21 65515200; Fax: +86 21 65492132; E-mail: chenghe@smmu.edu.cn
†These authors contributed equally to this work
The copyright line of this article was changed on 29 August 2017 after first online publication.

binding PRC2 to inhibit M2 polarization. GAS5 was shown to be involved in the regulation of EAE progression. Moreover, GAS5 inhibited human microglial M2 polarization and was highly expressed in the amoeboid-shaped microglia in MS lesions, suggesting a novel role for an lncRNA in MS pathogenesis.

# Results

### GAS5 expression is decreased in M2 polarized microglia

To find potential lncRNAs participating in the regulation of microglial polarization, we employed microarray screening. A total of 120 lncRNAs were differentially expressed (greater than twofold; $P < 0.05$) in IL-4-stimulated M2 microglia versus resting microglia (Fig 1A). Among them, GAS5 was significantly decreased in the M2 polarized microglia and was closely related with other differentially expressed molecules, including known macrophage polarization regulators such as TSC1 [21] (Fig 1B). This information prompted us to examine whether GAS5 is involved in the regulation of microglia polarization. First, we validated microarray results by qPCR and the results showed GAS5 was dramatically decreased in IL-4 stimulated primary cultured microglia (Fig 1C). We also examined the GAS5 level during microglial polarization induced by other factors. Macrophage colony-stimulating factor (M-CSF) has been reported to promote M2 polarization [22]. Consistent with the induction by

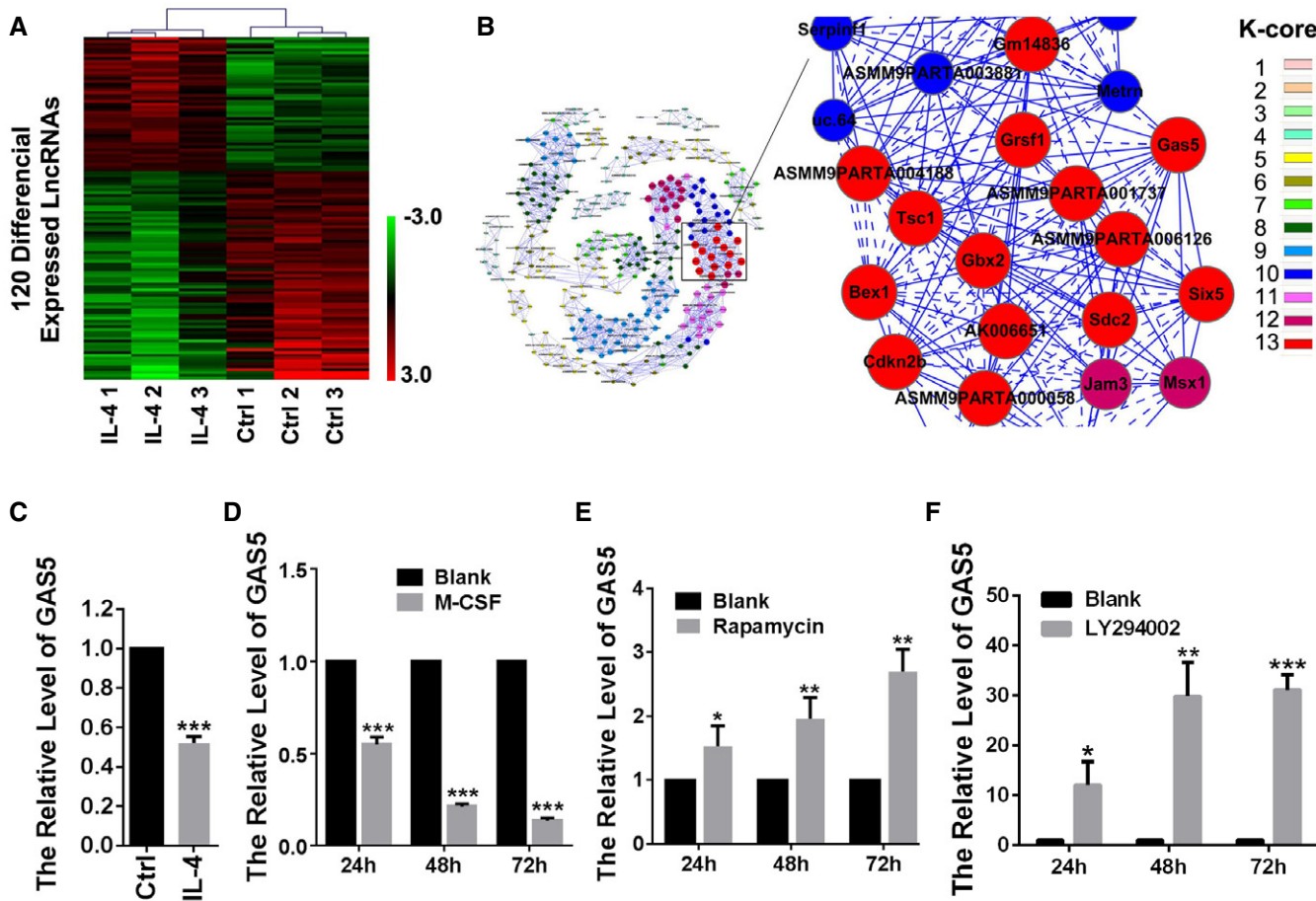

**Figure 1. GAS5 is downregulated in M2-polarized microglia.**

A   Hierarchical clustering analysis of 120 lncRNAs that were differentially expressed between microglia treated with IL-4 for 24 h and the untreated control (greater than twofold; $P < 0.05$). Expression values are represented in shades of red and green and indicate expression above and below the median expression value across all samples (log scale 2, from 3 to −3), respectively.

B   Co-expression network in IL-4-stimulated microglia. The co-expression network includes 1,627 connections among 319 genes that are correlated at $|r| > 0.99$. The color of the nodes is in accordance with the K-core number shown in the legend. Real lines between two nodes indicate positively correlated interactions between genes, and dashed lines indicate negatively correlated interactions. The enlarged box shows that GAS5 is a key in the network.

C   Quantitative PCR analysis of GAS5 in microglia treated with IL-4 for 24 h versus the untreated control, $n$ = 3 experiments.

D   Quantitative PCR analysis of GAS5 in M-CSF-treated microglia versus the untreated control, $n$ = 3 experiments.

E   Quantitative PCR analysis of GAS5 in rapamycin-treated microglia versus the untreated control, $n$ = 3 experiments.

F   Quantitative PCR analysis of GAS5 in LY294002-treated microglia versus the untreated control, $n$ = 3 experiments.

Data information: *$P < 0.05$, **$P < 0.01$, ***$P < 0.001$ (Student's *t*-test). Data are shown as the mean ± SD.

IL-4, we found that GAS5 was greatly decreased in M-CSF-stimulated primary cultured microglia (Fig 1D). Because rapamycin (mTOR inhibitor) has been shown to inhibit M2 polarization [23], we applied rapamycin to the cultured microglia and found that the GAS5 expression level was significantly increased (Fig 1E). The PI3K inhibitor LY294002, which also inhibits M2 polarization, showed a similar result (Fig 1F). These results indicate an inverse correlation between GAS5 expression and M2 polarization.

### GAS5 suppresses microglial M2 polarization and promotes M1 polarization

To investigate whether GAS5 directly regulated microglial polarization, we transfected primary cultured microglia with GAS5 overexpression (GAS5OE) or GAS5 interference (GAS5i) lentiviral vectors. qPCR analysis showed that GAS5 overexpression in microglia decreased the expression of the M2 markers Ym-1, Fizz-1, CD206, and IGF-1 and increased the expression of the M1 markers TNF-α and IL-1β (Fig 2A). Conversely, interference with GAS5 in microglia resulted in an increase in the expression of M2 markers and a decrease in the expression of M1 markers (Fig 2B, Appendix Fig S1A). ELISA and Western blotting showed similar changes in TNF-α, IL-1β, and IGF-1 protein expression in the supernatants or protein lysates from the microglia overexpressing GAS5 (MG$^{GAS5OE}$; Fig 2C and E) or the microglia with GAS5 interference (MG$^{GAS5i}$; Fig 2D and F). Overexpression or interference with GAS5 in peripheral monocytes resulted in a similar phenotype (Fig EV5C and D). Besides, to test the influence of GAS5 on the secretion of cytokines by microglia, we conducted co-culture experiments using the conditioned medium from GAS5-modified microglia *in vitro* and found OPC survival, and differentiation and neurite growth were significantly impaired in the GAS5OE group (Figs EV1 and EV2). These results indicate that GAS5 suppresses microglial M2 polarization and promotes M1 polarization.

### GAS5 expression in microglia during EAE

To elucidate the role of GAS5 in microglial functions *in vivo*, we induced EAE in C57BL/6 mice. First, we observed the GAS5 expression pattern in spinal cord sections from EAE mice 20 days post-immunization (20 dpi) by FISH. As shown in Fig 3A, GAS5 was expressed in Iba1$^+$ cells (microglia and infiltrating macrophages) and especially in the amoeba-shaped microglia/macrophages, which tended to be M1 polarized [24]. The ramification index is often used to quantitatively analyze the morphology of microglia/macrophages [25,26]. We found that the ramification index of GAS5$^+$ microglia/macrophages was significantly higher than the index in GAS5$^-$ cells (Fig 3C). The same went for TMEM119$^+$ microglia (Fig EV3). Further analysis revealed that more GAS5 was expressed in TNF-α$^+$ (M1 marker) cells than Arg-1$^+$ (M2 marker) cells (Fig 3E and F).

To quantify GAS5 expression in microglia during EAE, we isolated microglia from mice in the control group, during the peak stage of EAE (15 dpi) and during the chronic stage of EAE (40 dpi) (Fig 3G). The qPCR showed that the GAS5 level was higher in the EAE mice than the controls. The expression reached its highest point during the peak stage of EAE when inflammation was prominent (Fig 3H).

### GAS5-modified microglia regulate EAE progression

To elucidate the role of GAS5 in microglial functions *in vivo*, we used the adoptive transfer strategy and transplanted GAS5-modified microglia into the cerebral ventricles of EAE mice at 8 dpi. The results showed that the transplantation of MG$^{GAS5OE}$ exacerbated EAE (Fig 4A), whereas MG$^{GAS5i}$ alleviated EAE (Fig 4B). Moreover, H&E staining showed that the infiltration of inflammatory cells into the lumbar spinal cord of the EAE mice at 20 dpi was increased after MG$^{GAS5OE}$ transplantation and reduced after MG$^{GAS5i}$ transplantation (Fig 4C and D). Luxol fast blue (LFB) staining showed exacerbated myelin loss in the white matter after MG$^{GAS5OE}$ transplantation and attenuated myelin loss after MG$^{GAS5i}$ transplantation (Fig 4E and F).

The *in vitro* co-culture experiments and the *in vivo* transplantation work both showed protective effects of GAS5 interference in microglia. Next, we examined the effect of MG$^{GAS5i}$ application after EAE onset. MG$^{GAS5i}$ was transplanted into the cerebral ventricle of EAE mice at 20 dpi, and the lumbar spinal cord was collected at 40 dpi for histological analysis. A recovery in the EAE scores was detected after MG$^{GAS5i}$ transplantation (Fig 5A), with reduced inflammatory infiltrating cells (Fig 5B and D) and alleviated demyelination (Fig 5C and E). These results demonstrated that interfering with GAS5 in microglia was beneficial for EAE treatment.

### GAS5 interference in microglia promotes remyelination following lysolecithin-induced focal demyelination

Inflammation-induced demyelination is the core pathological feature of EAE. To avoid the influence of microglial transplantation on the immune reaction, we applied MG$^{GAS5i}$ in the lysolecithin-induced focal demyelination model to evaluate the effects on demyelination and remyelination. Focal demyelination was induced by injecting lysolecithin into the dorsal spinal cord white matter, and MG$^{GAS5i}$ was transplanted 3 days post-lysolecithin injection (dpl). The spinal cords around the injection points were isolated at 7, 11, and 15 dpl and subjected to LFB or immunohistofluorescence (IHF) staining. LFB staining showed increased remyelination at 11 and 15 dpl after MG$^{GAS5i}$ transplantation (Fig 6A and B). IHF staining further illustrated that more Olig2$^+$ oligodendrocyte lineage cells were accumulated in the lesion of MG$^{GAS5i}$-transplanted mice even at 7 dpl (Fig 6C and D). More significantly, CC1$^+$ cells (differentiated oligodendrocytes) were also increased within lesions in MG$^{GAS5i}$-transplanted animals at 7, 11, and 15 dpl (Fig 6C and E). Electron microscopy (EM) analysis revealed a higher percentage of remyelinated axons and a lower g-ratio within lesions after MG$^{GAS5i}$ transplantation at 15 dpl (Fig 6F–I), suggesting that interfering with GAS5 in microglia promoted remyelination.

### GAS5 suppresses IRF4 transcription by binding PRC2 and inhibiting microglial M2 polarization

To explore the mechanisms underlying the effect of GAS5 on microglia, we first conducted an RNA pull-down followed by silver staining after SDS–PAGE gel electrophoresis to identify proteins that specifically bound to GAS5. After incubation with the microglial protein lysate, no specific bands between the GAS5 mRNA and the

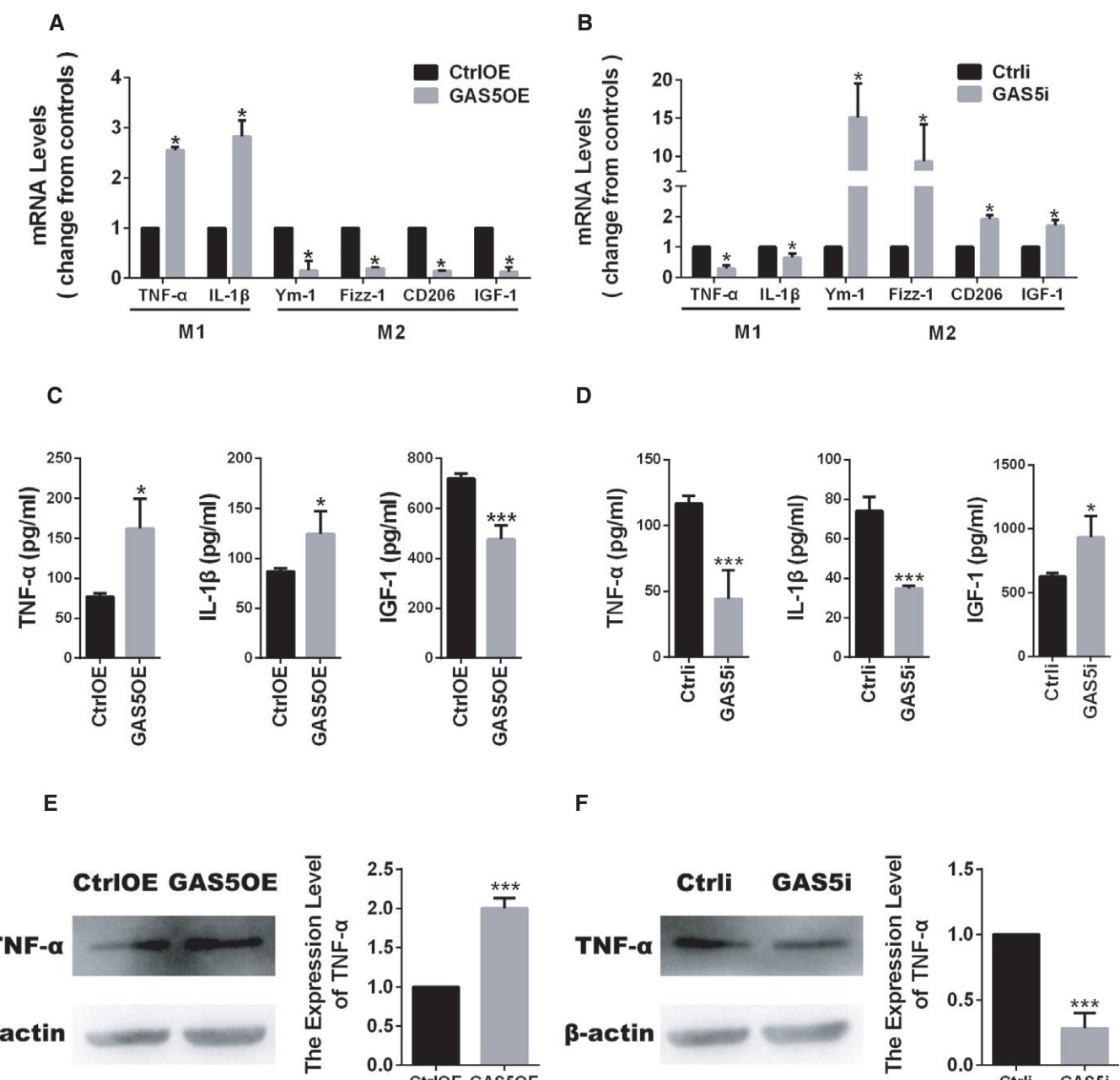

**Figure 2. GAS5 suppresses microglial M2 polarization and promotes M1 polarization.**

A, B   Quantitative PCR analysis of M1 and M2 markers in microglia transduced with the GAS5OE (A) or GAS5i (B) lentivirus vectors versus the control, $n \geq 3$ experiments.

C, D   ELISA analysis of TNF-α, IL-1β, and IGF-1 in culture supernatants of microglia transduced with the GAS5OE (C) or GAS5i (D) lentivirus vectors versus the control, $n \geq 3$ experiments.

E, F   Western blotting analysis of TNF-α in microglia transduced with the GAS5OE (E) or GAS5i (F) lentivirus vectors versus the control, $n \geq 3$ experiments.

Data information: *$P < 0.05$, ***$P < 0.001$ versus control (Student's *t*-test). Data are shown as the mean ± SD.

Source data are available online for this figure.

antisense control were detected by silver staining (Fig EV4A). GAS5 was reported to bind to the glucocorticoid receptor [27], but our RNA immunoprecipitation study showed that only a very small portion of GAS5 bound with this receptor (Fig EV4B), consistent with the report that the binding requires the addition of glucocorticoids [28]. Therefore, it is unlikely that GAS5 regulates microglial

polarization through glucocorticoid receptor signaling. Previous work showed that recruiting polycomb repressive complex 2 (PRC2) to the promoter region of target genes repressed transcription and was an important mechanism for long intergenic noncoding RNAs (lincRNAs) [15]. To test whether the lincRNA GAS5 interacted with PRC2, we performed an RNA immunoprecipitation assay with an

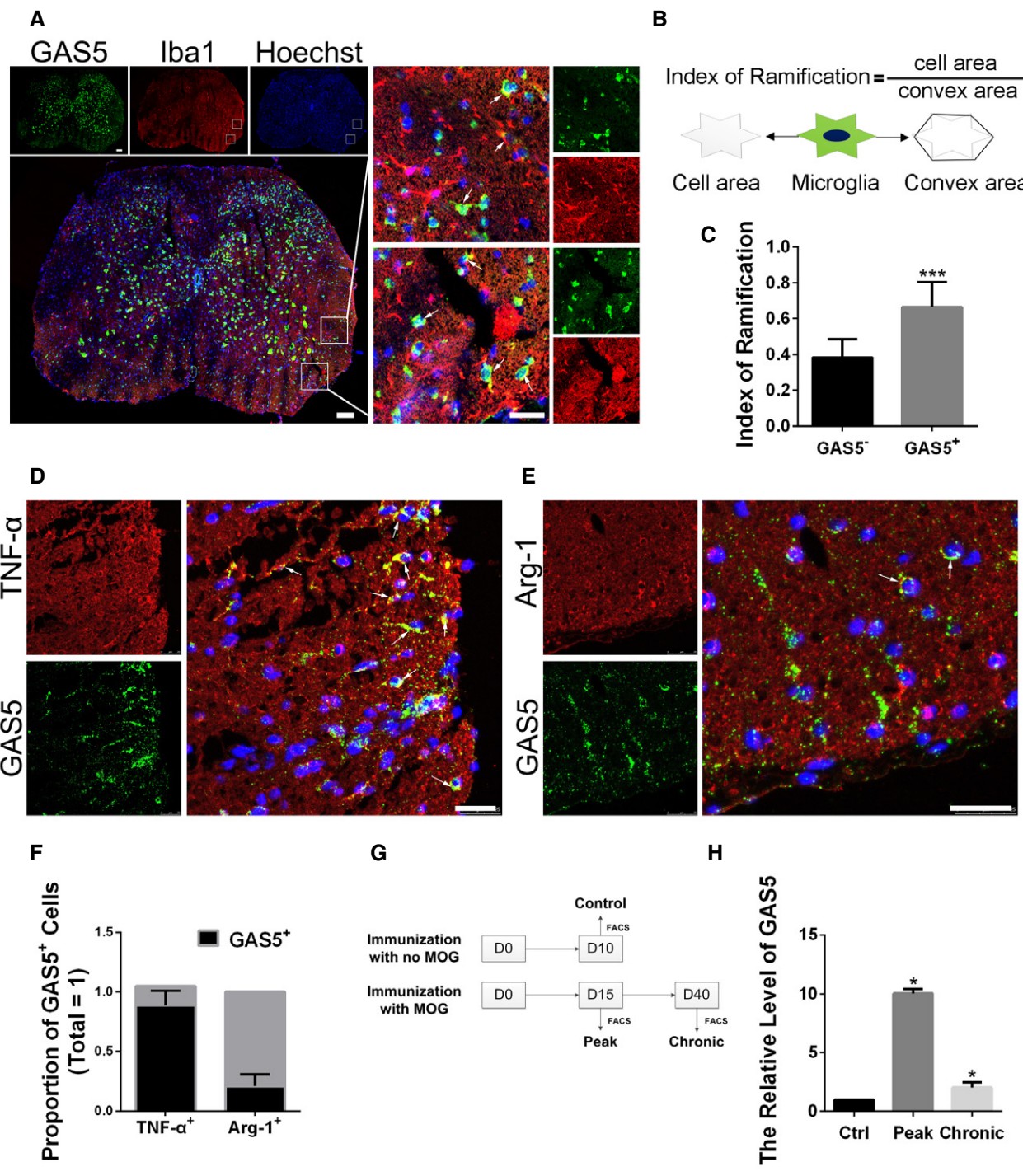

**Figure 3.  GAS5 is dynamically expressed in the microglia of EAE.**

A    Representative FISH analysis of GAS5 (green) co-stained with an anti-Iba1 antibody (red) in spinal cord sections from EAE mice.  Arrows indicate GAS5+Iba1+ cells.
Scale bars = 100 μm.
B    A schematic map of the calculation of the ramification index (RI).
C    Statistical analysis of the RI in Iba1+GAS5+ (n = 16) or Iba1+GAS5− (n = 17) cells.
D, E    Representative FISH analysis of GAS5 (green) co-stained with an anti-TNF-α (D) or anti-Arg-1 (E) antibody (red) in spinal cord sections from EAE mice. Arrows indicate GAS5+TNF-α+ cells in (D) and GAS5+Arg-1+ cells in (E). Scale bars = 25 μm.
F    Statistical analysis of the proportion of GAS5+ cells in (D, E), n ≥ 3 experiments.
G    A schematic map of the sorting of microglia in different phases of EAE and the control by FACS.
H    Quantitative PCR analysis of GAS5 in different phases of EAE versus the control, n = 3 experiments.

Data information: *$P < 0.05$, ***$P < 0.001$ (C, Student's t-test; H, one-way ANOVA with Dunnett's post hoc test). Data are shown as the mean ± SD.

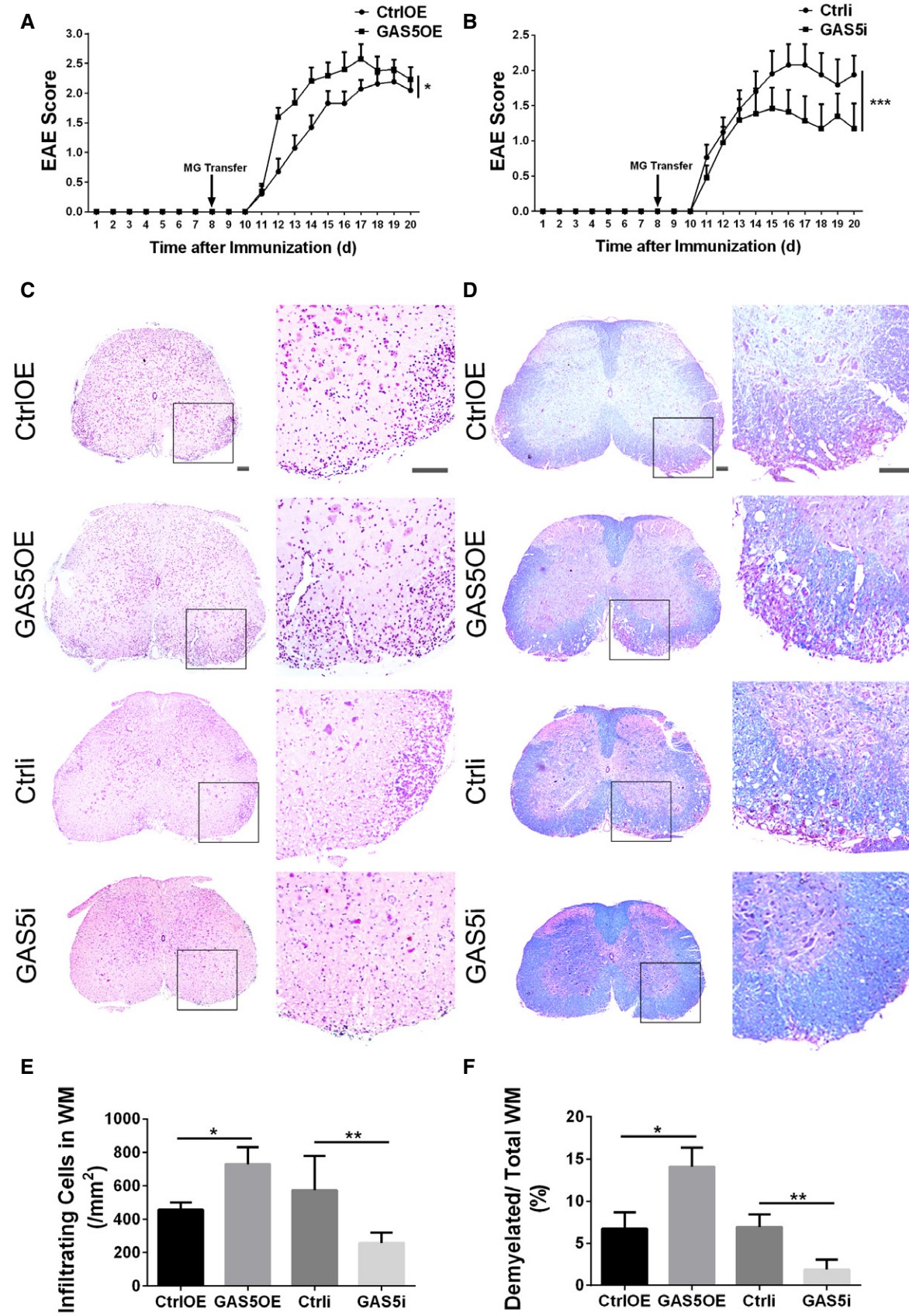

Figure 4.

**Figure 4.  GAS5-modified microglia regulate the progression of EAE.**

A, B Clinical scores of EAE in mice transplanted with MG$^{CtrlOE}$/MG$^{GAS5OE}$ (A) or MG$^{Ctrli}$/MG$^{GAS5i}$ (B) by lateral ventricle injection. CtrlOE, *n* = 18; GAS5OE, *n* = 18; Ctrli, *n* = 8; GAS5i, *n* = 10 mice.

C, D Representative spinal cord sections from EAE mice at day 20 after immunization after H&E staining (C) and Luxol fast blue staining (D). Scale bars = 100 μm.

E, F Quantification of infiltration in the white matter (WM) (E) and the percentage of demyelinated WM in total WM (F) in (C and D), *n* ≥ 3 mice per group.

Data information: *\*P* < 0.05, *\*\*P* < 0.01, *\*\*\*P* < 0.001 versus control (A, B, Mann–Whitney *U*-test; E, F, Student's *t*-test). Data are shown as the mean ± SEM (A, B) or mean ± SD (E, F).

antibody against enhancer of zeste homolog 2 (EZH2), which is the main catalytic subunit of PRC2. As shown in Fig 7A, a high proportion of GAS5 was observed to bind with EZH2. The RNA pull-down experiments confirmed the binding between GAS5 and EZH2 in reverse (Fig 7B). RIP experiments with another antibody against the main binding subunit of PRC2, RbAp48, showed a similar phenomenon (Fig EV4C). In contrast, RIP experiments with an anti-RING1A/B (subunits of PRC1) antibody showed no specific

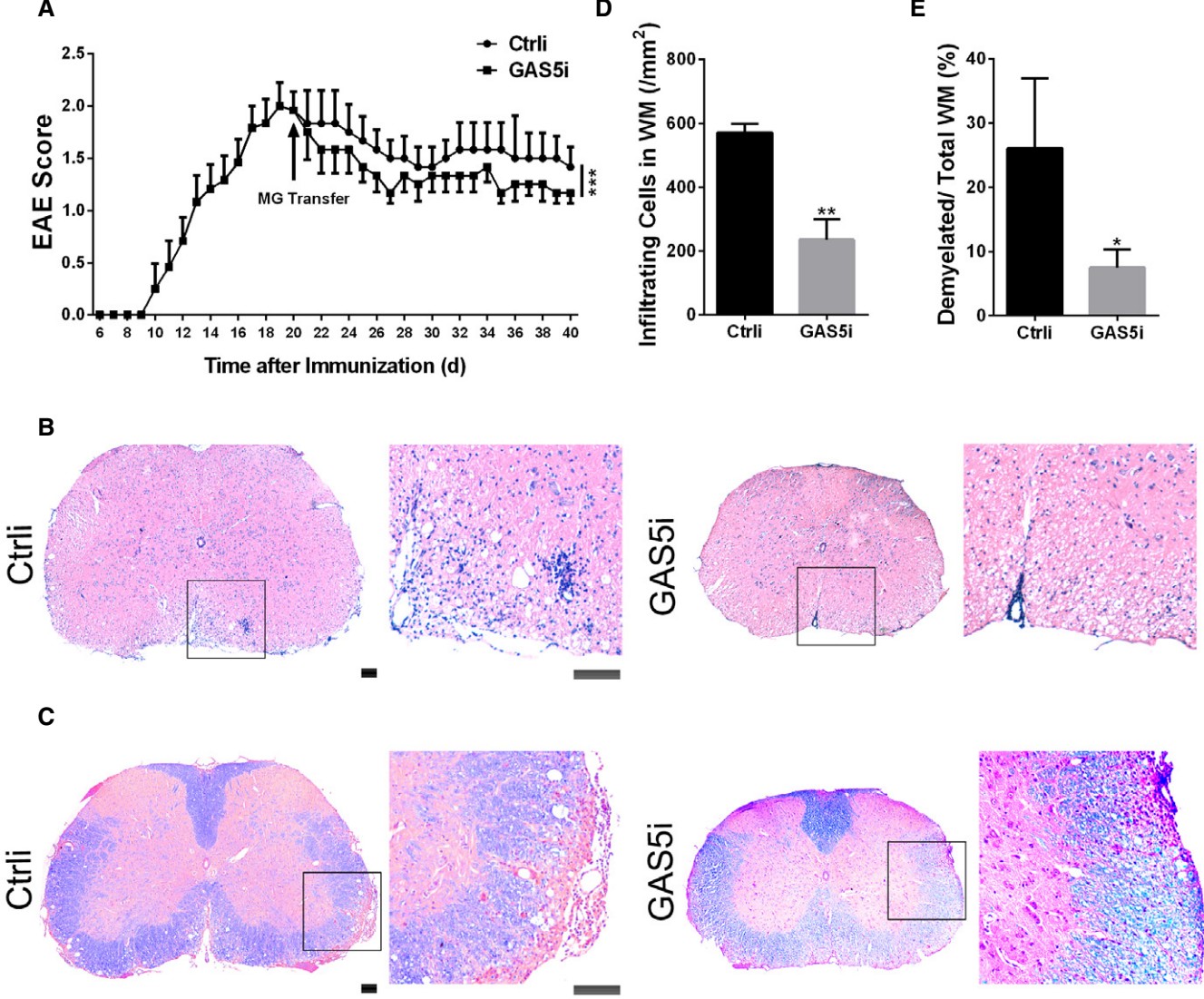

**Figure 5.  Interfering with GAS5 in microglia attenuates EAE progression.**

A Clinical scores of EAE in mice transplanted with MG$^{Ctrli}$/MG$^{GAS5i}$ by lateral ventricle injection, *n* = 6 mice per group.

B, C Representative spinal cord sections of H&E stained (B) and Luxol fast blue stained (C) EAE mice day 40 after immunization. Scale bars = 100 μm.

D, E Quantification of infiltration in the white matter (WM) (D) and the percentage of demyelinated WM in total WM (E) in (B and C), *n* = 3 mice per group.

Data information: *\*P* < 0.05, *\*\*P* < 0.01, *\*\*\*P* < 0.001 versus control (A, Mann–Whitney *U*-test; D, E, Student's *t*-test). Data are shown as the mean ± SEM (A) or mean ± SD (D, E).

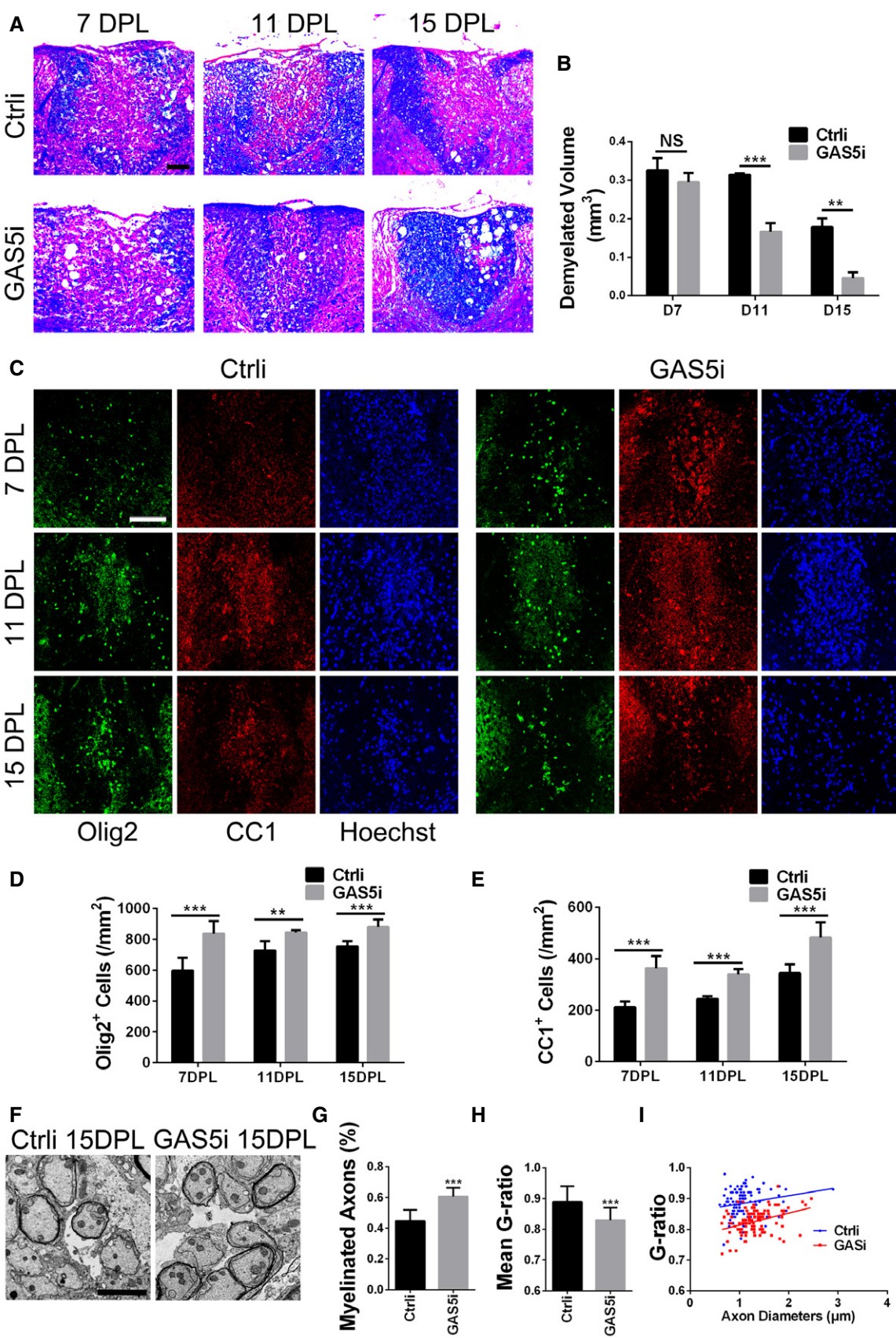

**Figure 6.**

**Figure 6.   Interfering with GAS5 in microglia promotes remyelination in a LPC-induced focal lesion model.**

A   Representative Luxol fast blue-stained dorsal column spinal cord sections from the MG$^{Ctrli}$/MG$^{GAS5i}$ groups at 7, 11, and 15 days post-LPC injection. Scale bar = 100 μm.
B   Quantitative analysis of the demyelinated lesion volume 5 mm around the epicenter of (A), n = 3 mice per group.
C   Representative anti-Olig2 (green) and anti-CC1 (red) immunohistofluorescence of dorsal column spinal cord sections from the MG$^{Ctrli}$/MG$^{GAS5i}$ groups at 7, 11, and 15 days post-LPC injection. Scale bar = 100 μm.
D, E   Statistical analysis of Olig2$^+$ (D) or CC1$^+$ (E) cells of (C), n ≥ 3 mice per group.
F   Representative electron micrographs of dorsal column spinal cord sections from the MG$^{Ctrli}$/MG$^{GAS5i}$ groups at 15 days post-LPC injection. Scale bar = 2 μm.
G   Quantification analysis of myelinated axons among total axons between MG$^{Ctrli}$ and MG$^{GAS5i}$ groups at 15 dpl, n = 3 mice per group.
H   Statistical analysis of g-ratio between MG$^{Ctrli}$ and MG$^{GAS5i}$ groups at 15 dpl.
I   Analysis of the myelinated axons showed a reduction of the g-ratio in spinal cords in the MG$^{GAS5i}$ group (red) compared with the MG$^{Ctrli}$ group (blue) at 15 dpl, n = 3 mice per group.

Data information: **$P < 0.01$, ***$P < 0.001$ versus control (Student's $t$-test). Data are shown as mean ± SD.

interaction (Fig EV4D). After knocking down EZH2 in microglia (Appendix Fig S1B), the qPCR showed increased M2 markers and decreased M1 markers, which further supported the notion that PRC2 participates in the regulation of microglial polarization (Fig 7C).

Next, we performed a ChIP assay to investigate whether the interaction between EZH2 and the promoter areas of genes involved in M2 polarization (i.e., PPARγ, STAT6, IRF4, and C/EBPβ) were affected by GAS5 [22]. The results showed binding between EZH2 and the IRF4 promoter was significantly enhanced in the microglia overexpressing GAS5 (Figs 7D and EV4E–G). ChIRP analysis further confirmed the binding between GAS5 and the promoter regions of IRF4 (Fig 7E), and ChIP using H3K27me3 antibody showed the methylation of H3K27 at the IRF4 promoter was changed accordingly (Fig 7F and G). Consistently, the qPCR and WB results showed that IRF4 expression was decreased in MG$^{GAS5OE}$ (Fig 7H and K) and increased in MG$^{GAS5i}$ (Fig 7I and L) and MG$^{EZH2i}$ (Fig 7J and M). These results showed that GAS5 suppressed IRF4 transcription by binding PRC2, thereby inhibiting microglial M2 polarization. In addition, we analyzed another epigenetic histone mark by performing ChIP assays using anti-H3K4me3 antibody. The results show that H3K4me3 was decreased at the IRF4 promoter in MG$^{GAS5OE}$, but did not change in MG$^{GAS5i}$ (Appendix Fig S2), which is in accordance with the inhibited transcription observed upon GAS5 overexpression.

### GAS5 inhibits human microglial M2 polarization and is differentially expressed in MS lesions

LncRNAs usually lack conservatism among species, which limits the universality of functions. However, GAS5 has a relatively high

homology between mice and humans. Therefore, we hypothesized that human GAS5 may have similar functions. We constructed a vector encoding human GAS5 and transfected human microglia using electroporation. The qPCR showed that the M2 markers were decreased and M1 markers were increased (Fig 8A). Following a homo-transfer strategy to study the function of an lncRNA across species [29], we transfected human GAS5 into mouse microglia using electroporation; the qPCR showed an effect similar to mouse GAS5 in microglial polarization (Fig 8B). These results showed that the function of GAS5 in microglial polarization in humans and mice was conserved.

We also examined the expression of GAS5 in microglia from MS patients. Using brain tissues from MS patients kindly provided by the Netherlands Brain Bank (NBB), we performed FISH and found that more GAS5 was present in the amoeba-shaped microglia/macrophages, which was in agreement with the finding from EAE (Fig 8C). The ramification index of GAS5$^+$ microglia/macrophages was significantly higher than the index for GAS5$^-$ cells (Fig 8D). These results strongly suggested that GAS5 might be involved in the regulation of microglial functions in MS.

## Discussion

From transcriptional "noise" to functional participants, the roles of lncRNAs in neurological disorders are becoming recognized [15]. GAS5 is a functional lncRNA identified decades ago that has been shown to be abnormally expressed in many types of tumors and to inhibit T-cell proliferation [18,20]. In the present study, we demonstrate a novel role for GAS5 in microglial polarization. As the local macrophages, microglia are the major source of M2 cells in the CNS

**Figure 7.   GAS5 suppresses IRF4 transcription by binding PRC2 and inhibiting microglial M2 polarization.**

A   RNA IP analysis of the binding between EZH2 and GAS5, n = 3 experiments.
B   RNA pull-down analysis of the binding between EZH2 and GAS5.
C   Quantitative PCR analysis of M1 and M2 markers in microglia transduced with the EZH2i lentivirus versus the control, n ≥ 4 experiments.
D   ChIP analysis of microglia transduced with the CtrlOE or GAS5OE lentivirus. A relatively high enrichment was detected on the IRF4 promoter regions in MG$^{GAS5OE}$ versus the control using the anti-EZH2 antibody, n = 3 experiments.
E   ChIRP analysis of the binding between the IRF4 promoter and GAS5, n = 3 experiments. IGF1 served as a negative control.
F, G   ChIP analysis of microglia transduced with the CtrlOE/GAS5OE (F) or Ctrli/GAS5i (G) lentivirus on the IRF4 promoter regions using the anti-H3K27me3 antibody, n = 3 experiments.
H–J   Quantitative PCR analysis of IRF4 in microglia transduced with the GAS5OE (H), GAS5i (I) or EZH2i (J) lentivirus, n ≥ 3 experiments per group.
K–M   Western blotting analysis of IRF4 in microglia transduced with the GAS5OE (K), GAS5i (L), or EZH2i (M) lentivirus, n ≥ 3 experiments per group.

Data information: *$P < 0.05$, **$P < 0.01$, ***$P < 0.001$ versus control or IgG (A) (Student's $t$-test). Data are shown as the mean ± SD.
Source data are available online for this figure.

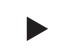

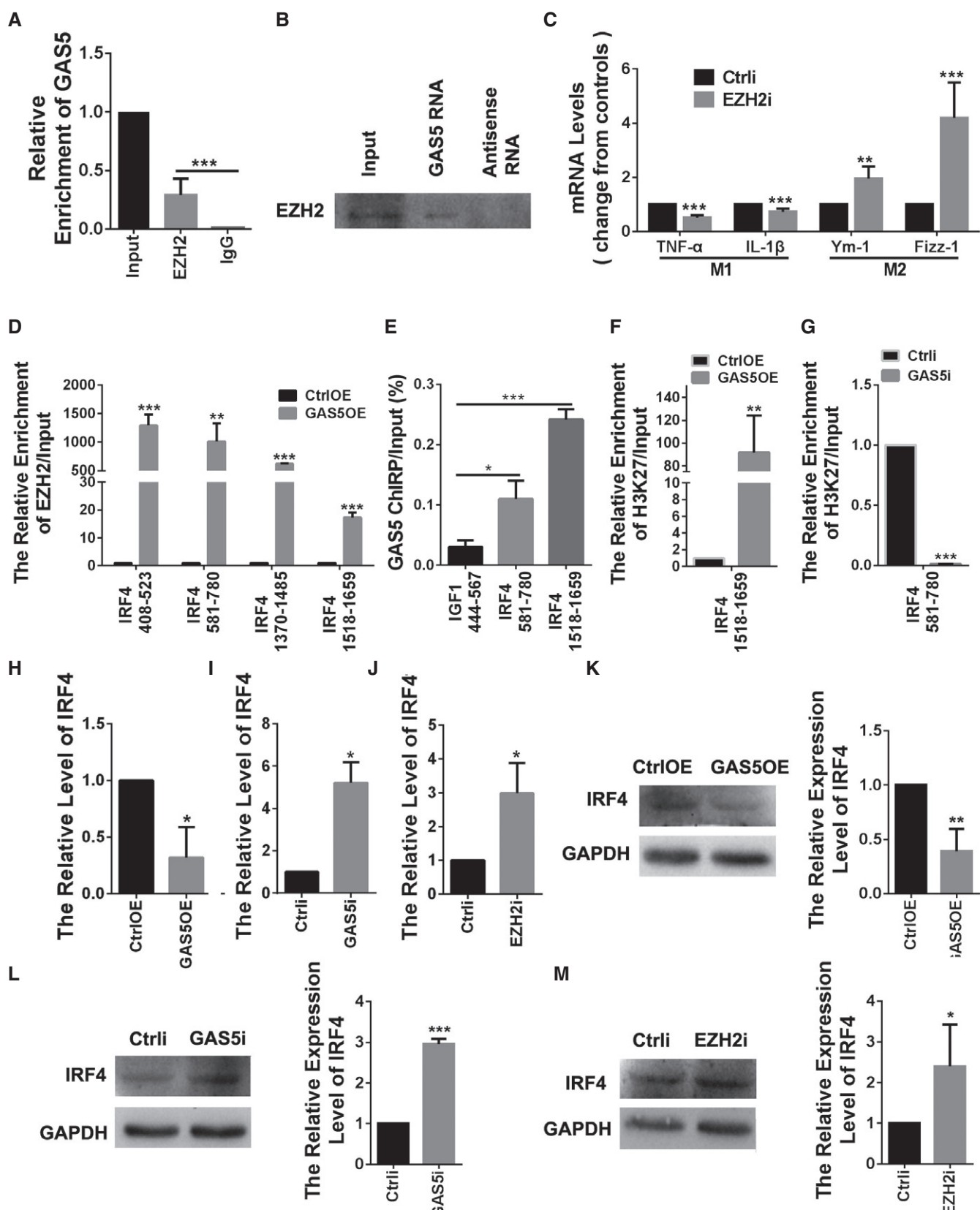

Figure 7.

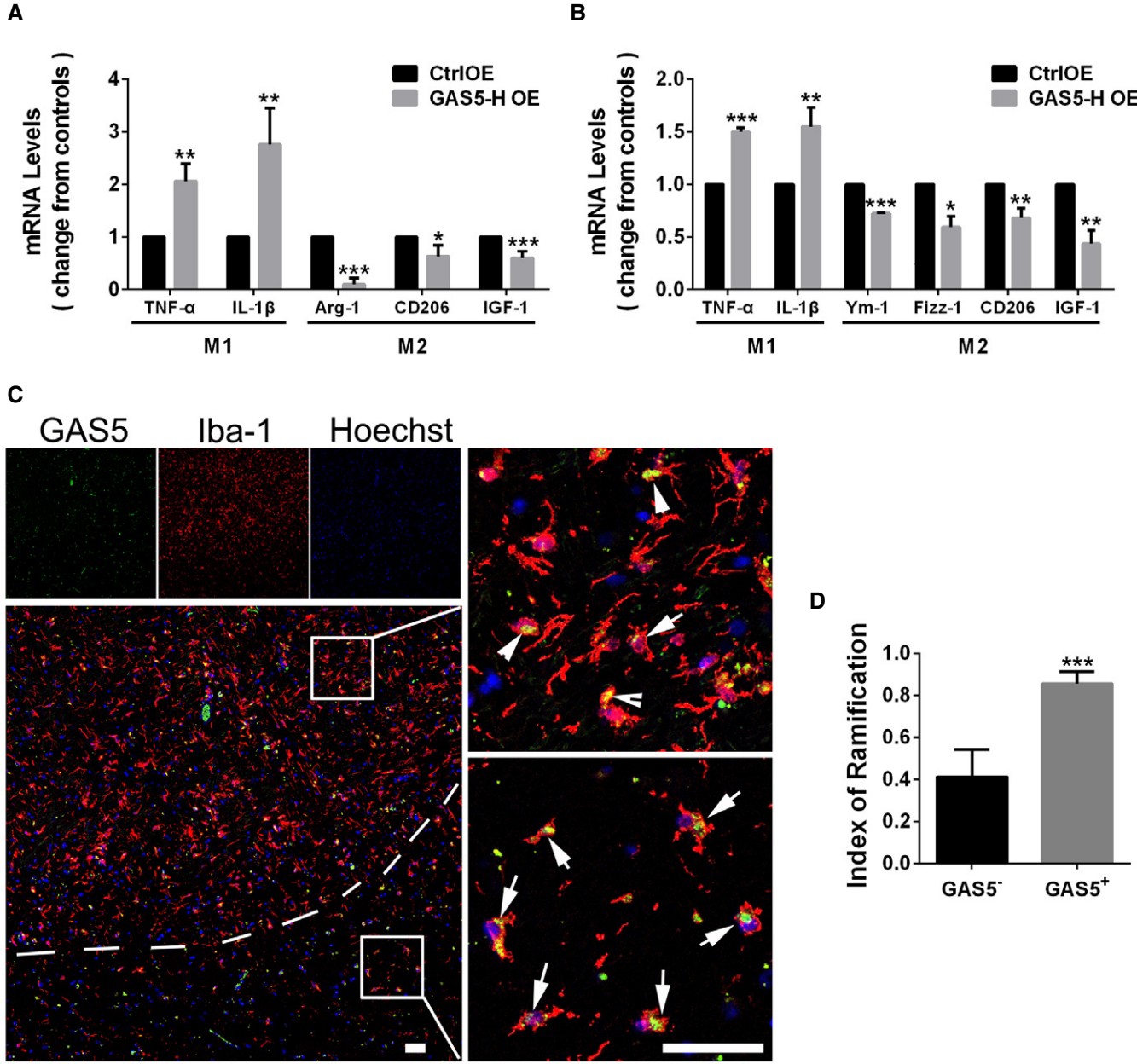

**Figure 8. GAS5 inhibits human microglial M2 polarization and is differentially expressed in MS lesions.**

A  Quantitative PCR analysis of M1 and M2 markers in human microglia nucleo-transfected with the human GAS5 plasmid versus the control, $n \geq 3$ experiments.

B  Quantitative PCR analysis of M1 and M2 markers in mouse microglia nucleo-transfected with the human GAS5 plasmid versus the control, $n = 3$ experiments.

C  Representative FISH analysis of GAS5 (green) co-stained with the anti-Iba1 antibody (red) in brain slices from MS patients. Arrows indicate GAS5[+]Iba1[+] cells. Scale bar = 50 μm.

D  Statistical analysis of the RI of Iba1[+]GAS5[+] or Iba1[+]GAS5[−] cells in brain slices from MS patients, $n = 6$.

Data information: *$P < 0.05$, **$P < 0.01$, ***$P < 0.001$ (Student's $t$-test). Data are shown as the mean ± SD.

[30]. The lack of M2 polarized microglia/macrophages impairs remyelination in MS and other demyelinating diseases [9,10]. We found that GAS5 modification in microglia affected the severity of EAE. Interfering with GAS5 in microglia attenuated EAE progression and promoted remyelination in a LPC model. Moreover, we found that GAS5 exhibited higher expression in amoeba-shaped microglia/macrophages in the brain slices of MS patients and mice with EAE.

We also demonstrated that the function of GAS5 in microglial polarization was conserved in humans and mice. Our work suggested that GAS5 was a promising target for the treatment of demyelinating diseases such as MS.

Currently, little is known about GAS5 signaling, although GAS5 has been shown to act as a "decoy" to combine with the DNA-binding domain of glucocorticoid receptor and block its interactions

with downstream targets [27]. However, GAS only interacts with ligand-activated GR and only a very small portion of GAS5 binding with GR at physiologic level of circulating cortisol, while no binding exists without GR ligand. GAS5 was found regulating microglial polarization under serum-free conditions, which indicates the role of GR in the regulation of microglial polarization by GAS5 is ignorable. In the present study, we showed that GAS5 suppressed IRF4 transcription by binding with the major intracellular inhibitory compound PRC2. IRF4 is one of the most important transcription factors involved in microglial M2 polarization [22]. A previous study showed that Jmjd3 increased IRF4 transcription by enhancing the demethylation of histone 3 Lys27 in the IRF4 promoter region to promote macrophage M2 polarization [31]. In this study, GAS5 interacted with PRC2 by binding with EZH2, which was the main catalytic subunit of PRC2. Importantly, components of the polycomb repressive complex 1 (PRC1) showed no interaction with GAS5, supporting a specific role for PRC2 in recruiting lncRNAs. Furthermore, GAS5 overexpression enhanced the interaction between EZH2 and the IRF4 promoter and decreased IRF4 expression. Consistently, interference with GAS5 or EZH2 in microglia increased IRF4 expression. These results suggest that GAS5 binds PRC2 to suppress IRF4 transcription and inhibit microglial M2 polarization. Further, the active epigenetic mark H3K4me3 was reduced in MG$^{GAS5OE}$ at the IRF4 promoter, but showed little changes in MG$^{GAS5i}$, confirming the reduced expression of IRF4 observed upon GAS5 overexpression, and indicating that further regulative chromatin modifying mechanisms act during this process.

In contrast to macrophage M1 polarization, the mechanism underlying M2 polarization is not fully understood [22,32]. mTOR activation has been shown to promote macrophage M2 polarization [23]. GAS5 is a member of the 5′-terminal oligopyrimidine tract (5′-TOP) family that is controlled by mTOR and has been shown to be upregulated by rapamycin (an mTOR inhibitor) in prostate cancer and T cells [33,34]. Consistent with these reports, we found that rapamycin treatment dramatically increased the GAS5 level in microglia, suggesting that mTOR was upstream of GAS5 in the regulation of microglial polarization. In the present study, we also found that GAS5 expression was decreased after IL-4 and M-CSF stimulation, both of which were reported to promote M2 polarization. IL-4 promotes macrophage M2 polarization through PI3K and induces the expression of IRF4 [35,36], whereas M-CSF can also activate PI3K and promote macrophage M2 polarization by regulating IRF4 [22]. Because PI3K is upstream of mTOR and GAS5 controls IRF4 expression [37], it is likely that an IL-4/M-CSF-PI3K-mTOR-GAS5-IRF4 pathway exists. This finding also explains why GAS5 expression decreased under IL-4 and M-CSF stimulation.

A further interesting point is that GAS5 is closely related to cell proliferation [20]. We found that microglia from aged mice sorted by flow cytometry showed higher GAS5 expression (Fig EV5A and B). It is possible that this epigenetic regulation may account for the obstacle in microglial M2 polarization and the relevant failure in remyelination in aged mice [14,38]. Recently, Doring et al [39] reported that promoting macrophage/microglial proliferation with amphotericin B and M-CSF benefited remyelination. It would be interesting to test whether GAS5 is involved in this process as a more direct target that regulates the function of aged microglia.

In summary, the data presented here demonstrated a novel role for GAS5 in microglial polarization and the pathogenesis of demyelinating diseases. These results suggest the potential therapeutic benefit of targeting GAS5 for the treatment of neurological disorders associated with microglial polarization, including MS, Alzheimer's disease, and Parkinson's disease.

## Materials and Methods

### Animal experiments

The animal experiments in this study were performed in adherence with the National Institutes of Health *Guidelines on the Care and Use of Laboratory Animals* and were approved by the Animal Experimentation Ethics Committee of the Second Military Medical University. All animals were purchased from Shanghai Slac Laboratory Animal Co., Ltd (Shanghai, China). The detailed EAE induction methods were described previously [13]. Briefly, female C57BL/6 mice (8–10 weeks of age) were injected with MOG$_{35-55}$ (GL Biochem, Shanghai) in complete Freund's adjuvant at 3 points in the back. The immunization day was defined as day 0. Pertussis toxin (516561, Calbiochem-EMD Chemicals) in PBS was administered intraperitoneally on days 0 and 2. Clinical manifestations were examined daily by a blinded observer.

For the lysolecithin model, spinal cord lesions were induced as previously described [40]. Briefly, female C57BL/6 mice (6–8 weeks) were deeply anesthetized with 3% chloral hydrate and a laminectomy was performed. After fixing the spine, 1 μl of 1% lysolecithin (62962, Sigma-Aldrich) in a 0.9% sodium chloride solution was injected into the dorsal funiculus at the level of the T11–T12 vertebrae. The day of lysolecithin injection was designated day 0 (0 dpl). The spinal cord around the injection point was isolated and cut into serial cryosections. The demyelinated lesion volume 5 mm around the epicenter was calculated based on the equation

$$V = \sum \text{demyelinated lesion area} \times \text{width of section.}$$

For transplantation of microglia in the EAE model, a small hole was drilled into the mouse skull and $5 \times 10^5$ cells in 10 μl were inserted into the right lateral ventricle with a microinjector. For transplantation of microglia in the lysolecithin-induced focal lesion model, $5 \times 10^5$ cells in 2 μl were injected in the area near the lysolecithin injection point using the above paradigm at 3 dpl. For both models, microglia were injected at a flow rate of 1 μl/min, and all injections were performed with a stereotaxic apparatus.

### Cells cultures

Detailed methods for the primary culture and purification of microglia and OPCs were previously described [13,41]. Briefly, P0 C57BL/6 mice were sacrificed to acquire mixed cerebrum glial cell cultures containing microglia. After 10–14 days of growth in DMEM/F12 medium (Invitrogen) containing 10% fetal bovine serum (FBS, GBICO), the cultures were shaken for 6 h at 180 rpm at 37°C to collect the microglia. For M2 polarization, the microglia were stimulated with IL-4

(10 ng/ml, R&D). To elucidate the role of mTOR in GAS5 expression, the microglia were stimulated with rapamycin (20 nM, LC laboratories). To obtain OPCs, P0 Sprague Dawley rats (SD rats) were sacrificed to perform mixed cortical glial cell cultures. After 8–10 days of growth in DMEM containing 10% FBS, the cultures were shaken for 1 h at 180 rpm at 37°C to remove the microglia. After replacing the medium, the cultures were shaken for another 14 h at 200 rpm at 37°C to collect the OPCs.

Primary cultured neurons were obtained from the cortices of P0 C57BL/6 mice as previously described [13]. After dissection and digestion, the cortical neurons were suspended and seeded at a density between 5,000 and 50,000 cells/cm$^2$ onto poly-D-lysine-coated coverslips. The medium was replaced with neurobasal medium (GIBCO) supplemented with 2% B27 (GIBCO) after 4 h. OPCs and neurons were co-cultured with microglia conditioned media at a 1:1 ratio.

Primary human microglia (HM1900) were purchased from ScienCell Research Laboratories (Carlsbad) and cultured in DMEM/F12 medium containing 10% FBS and 10 ng/ml macrophage–colony stimulation factor (M-CSF, Sigma-Aldrich).

### Microarray analysis

The microarray assay was performed using the Mouse LncRNA Array V2.0 (Arraystar). Quantile normalization and subsequent data processing were performed using the GeneSpring GX v11.5.1 software package (Agilent Technologies). The threshold for differentially expressed genes was fold change ≥ 2 and *P*-value ≤ 0.05. Hierarchical clustering was performed using the TM-MEV software (MultiExperiment Viewer).

The co-expression analysis was performed with the assistance of Novel Bioinformatics Company. Briefly, the adjacency between two genes was defined as a power of the Pearson correlation between the corresponding gene expression profiles according to the normalized signal intensity. When creating the visual representation, only the strongest correlations (0.99 or greater) were drawn in these renderings. In the gene co-expression networks, each gene corresponds to a node, and two genes are connected by an edge (either positive or negative). A degree is defined as the number of directly linked neighbors. A k-core of a network or a graph is a subgraph in which all nodes are connected to at least k other nodes in the subgraph [42].

### Lentivirus transduction and electroporation

The mouse GAS5 sequence (IMAGE clone 3585621, Thermo) was ligated into the GV303 plasmid (GeneChem, Shanghai). The siRNAs for mouse GAS5 or EZH2 were ligated into the GV113 plasmid (GeneChem). The sequences for GAS5 siRNA were 5′-GCGAGCG CAATGTAAGCAA-3′ and 5′-CCTCTGTGATGGGACATCT-3′, and the sequences for EZH2 siRNA were 5′-AAGCACAATGCAACACCAA-3′ and 5′-GCTTTGGACAACAAGCCTT-3′. The titers of concentrated viral particles ranged between 3 × 10$^8$ and 1 × 10$^9$ transducing units/ml. Lentiviral particles were added to cultured microglia at a multiplicity of infection (MOI) = 10, and the supernatant was changed 12 h after infection.

The human GAS5 sequence was cloned and ligated into the pCDH dual-promoter plasmid (CD513B-1, SBI). Mouse microglia

and human microglia were collected and transfected using nucleofection (programs Y-001 and Y-010, respectively; Amaxa).

### Flow cytometry

The isolation of microglia from EAE mice was performed as previously described [13]. Briefly, EAE mice were sacrificed, and the hindbrain and spinal cord were collected for homogenization and digestion. After gradient centrifugation using Percoll (17-0891-09, GE Healthcare), the majority of the mononuclear cells were collected and stained with PE-CD45 (BD Biosciences) and FITC-CD11b (BD Biosciences) antibodies after incubation with a CD16/CD32 (BD Biosciences) antibody as a block. CD11b$^+$CD45$^L$ cells were sorted on a Moflo XDP (Beckman Coulter) and assessed.

For apoptosis analysis, OPCs were stained with FITC-Annexin-V and 7-AAD using a Cell Apoptosis Analysis Kit (Sungene, Tianjin) according to the manufacturer's instructions. Annexin-V$^+$7-AAD$^-$ cells were considered cells in early apoptosis, whereas Annexin-V$^+$7-AAD$^+$ cells were considered cells in late apoptosis.

### RNA isolation and quantitative PCR

Total RNA was extracted with the TRIzol reagent (Invitrogen), and first-strand cDNA was synthesized using a RevertAid First Strand cDNA Synthesis Kit (Thermo Scientific Fermentas) according to the manufacturer's instructions. Quantitative PCR (qPCR) was performed on a LightCycler 96 apparatus (Roche) using the SYBR Green Real-time PCR Master Mix (TOYOBO). Gene expression was normalized to a standard housekeeping gene (GAPDH) using the $\Delta\Delta C_T$ method. Primers were listed in Appendix Tables S2 and S3.

### Western blot and ELISA

Primary cell cultures were homogenized in NP40 buffer (Beyotime, China) supplemented with a protease inhibitor cocktail (Roche). The cell lysates were subjected to Western blotting using the anti-IRF4 (1:500, sc-6059, Santa Cruz), anti-TNF-α (1:500, AF-510-NA, R&D), anti-MBP (1:500, MAB382, Chemicon), anti-EZH2 (1:500, ab3748, Abcam), and HRP-conjugated anti-GAPDH (Kangcheng, Shanghai, China) antibodies. The protein bands were analyzed using Image Lab analysis (Bio-Rad).

The TNF-α, IL-1β, and IGF-1 levels in the supernatants were detected using an ELISA according to the manufacturer's instructions (EK2822, EK201B2, Multi Sciences, Hangzhou, China; MG100, R&D). The sample concentrations were calculated using an equation generated from a standard curve.

### Fluorescence *in situ* hybridization (FISH) and immunocytofluorescence staining (ICF)

An approximately 500-bp mouse or human GAS5 sequence (Appendix Table S1) was cloned and ligated into the pEASY-T5 plasmid (Transgen, Beijing, China) containing the T3 and T7 promoters, respectively. Antisense and sense digoxigenin-labeled cRNA probes were synthesized with a DIG-RNA Labeling Kit (12430721, Roche) with linearized templates. *In situ* hybridization was performed according to the previous report [43].

For ICF, cells or tissue sections were fixed, permeabilized, and incubated with primary antibodies (MBP, MAB382, Chemicon; MAP2, M9942, Sigma-Aldrich; TNF-α, AF-510-NA, R&D; Arg-1, 610708, BD; Olig2, AB9610, Millipore; CC1, ab16794, Abcam; TMEM119, ab209064, Abcam; and Iba1, 019-19471, Wako) overnight at 4°C, followed by 2 h of incubation with TRITC- or FITC-conjugated secondary antibodies (Jackson Immuno Research). Then, the samples were counterstained with Hoechst 33342 (Sigma-Aldrich). Fluorescence images were captured using fluorescence microscopy (DXM, Nikon) or confocal microscopy (Leica) and analyzed using Image-Pro Plus (Media Cybernetics).

### TUNEL assays

TUNEL assay was carried out using the *In Situ* Cell Death Detection Kit, TMR red (12156, Roche) according to the manufacturer's instructions. Briefly, the specimens were fixed, permeabilized, and subsequently incubated with the TUNEL reaction solution mixture in a humidified 37°C chamber for 1 h. The cell nuclei were labeled with Hoechst 33342. The percentage of TUNEL-labeled cells versus Hoechst-labeled cells was calculated.

### Histological staining

Lumbar spinal cords from EAE or lysolecithin-injected mice were isolated and cut into cryosections or paraffin sections. Every 7[th] (cryosection) or 15[th] (paraffin section) section was collected from each animal. The sections were stained with hematoxylin and eosin (H&E) solution or Luxol fast blue (LFB) and periodic acid schiff (PAS). The inflammatory infiltrating cells for H&E and the demyelination area for lysolecithin were calculated by blinded independent readers.

### Transmission electron microscopy (EM)

Briefly, mice were sacrificed and transcardially perfused with 4% PFA. Then, the dorsal column of spinal cords was isolated and fixed in 2.5% glutaraldehyde for 2 h and post-fixed in 1% osmium tetroxide for 45 min before being dehydrated and embedded in araldite resin. Ultrathin sections (60 nm) were stained in uranyl acetate and lead citrate. Samples were visualized using a transmission electron microscope (Hitachi H-7650, Japan) at 100 KV.

### RNA immunoprecipitation

The RNA immunoprecipitation (RIP) experiments were performed using the Magna RIP RNA-Binding Protein Immunoprecipitation Kit (17-700, Millipore) following the manufacturer's instructions. After incubation of the microglial lysate with the anti-EZH2 (ab3748, Abcam) or anti-GR (ab3580, Abcam) or anti-RbAp48 (ab1765, Abcam) or anti-RING1A/B (ab32644, Abcam) antibody, the co-precipitated RNA was detected by qPCR. The ratio of RNA binding with a specific antibody versus the total RNA was calculated.

### RNA pull-down

The RNA pull-down experiments were performed using the Magnetic RNA-Protein Pull-Down Kit (20164, Thermo) following the manufacturer's instructions. The retrieved proteins were detected by silver staining after SDS–PAGE gel electrophoresis or Western blotting with an anti-EZH2 antibody (ab3748, Abcam).

### ChIP and chromatin isolation by RNA purification (CHIRP)

ChIP was performed using an EZ ChIP™ Chromatin Immunoprecipitation Kit (17-371, Millipore) following the manufacturer's protocol. Quantitative PCR was conducted to detect DNA fragments binding with EZH2 or H3K27me3 (17-622, Millipore) or H3K4me3 (17-614, millipore). Primers were designed to detect a ~2,000-bp area before the transcriptional start site. The ratio of DNA binding with EZH2 versus total DNA was calculated and compared between groups.

CHIRP was performed according to a previous report [44]. Briefly, biotin-labeled GAS5 cRNA probes as used in FISH were transcribed with the Biotin RNA Labeling Mix (Roche) and T7 RNA polymerase (Thermo) *in vitro*, treated with RNase-free DNase I (Thermo), and purified with the RNA Purification Kit (Transgen). Primary cultured microglia were collected and sonicated as in ChIP with adding RNase inhibitors. Then, the hybridization was performed with biotinylated cRNA probes at 60°C overnight in hybridization solution. After adding streptavidin magnetic beads (Roche) and incubating for 30 min at 37°C, beads were washed and bound chromatin was isolated as in ChIP. Quantitative PCR was conducted to detect DNA fragments binding with GAS5 and the ratio of DNA binding with GAS5 versus total DNA was calculated.

### Statistics

A two-tailed Student's *t*-test was applied for the statistical comparison of two groups, and a one-way ANOVA with Dunnett's *post hoc* test was used for multiple groups. The EAE model was analyzed using the nonparametric Mann–Whitney *U*-test. The data were presented as the mean ± SD unless otherwise indicated. A $P < 0.05$ was considered statistically significant.

### Data deposition

All microarray data have been deposited in the Gene Expression Omnibus (GEO GSE99216).

Expanded View for this article is available online.

### Acknowledgements
Research in the author's laboratory is supported by the National Natural Science Foundation of China (81461138035, 31130024, 81371326, 31371068, 31571066) and core support grant from the Wellcome Trust and MRC to the Wellcome Trust – Medical Research Council Cambridge Stem Cell Institute.

### Author contributions
LC and CH designed the experiments and contributed to the writing of the manuscript. DS, ZY, XF, ML, YP, QS, DW, XZ, AH, and ZX conducted experiments. CZ and RJMF analyzed and interpreted results and contributed to the writing of the manuscript.

### Conflict of interest
The authors declare that they have no conflict of interest.

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
