## [Review Process File · EMBO Reports]

Manuscript EMBO-2016-43668

LncRNA GAS5 inhibits microglial M2 polarization and exacerbates demyelination

Dingya Sun, Zhongwang Yu, Xue Fang, Mingdong Liu, Yingyan Pu, Qi Shao, Dan Wang, Xiaolin Zhao, Aijun Huang, Zhenghua Xiang, Chao Zhao, Robin J. M. Franklin, Li Cao, and Cheng He

Corresponding authors: Li Cao and Cheng He, Second Military Medical University

Review timeline:

Submission date:	14 November 2016
Editorial Decision:	21 December 2016
Revision received:	22 April 2017
Editorial Decision:	17 May 2017
Revision received:	05 June 2017
Editorial Decision:	09 June 2017
Revision received:	16 July 2017
Accepted:	20 July 2017

Editor: Achim Breiling

Transaction Report:

1st Editorial Decision

21 December 2016

Thank you for the submission of your research manuscript to EMBO reports. We have now received reports from the three referees that were asked to evaluate your study, which can be found at the end of this email.

As you will see, all referees acknowledge the potential interest of the findings. However, all have raised a number of concerns and suggestions to improve the manuscript, or to strengthen the data and the conclusions drawn, which need to be addressed during a revision. As the reports are below, I will not detail them here.

Further, we think that the data on the EZH2-GAS5 interaction and IRF4 regulation are rather preliminary and need further strengthening. Other PRC1/2 proteins should be tested as in Fig. 9a and 9b to show a specific interaction with EZH2. As it is claimed that recruitment of PRC2 complexes by GAS5 to IRF4 leads to repression, H3K27me3 and H3K4me3 patterns need to be analyzed by Chip in Fig. 9a and correlated with the activity state of IRF4 (under normal conditions and upon GAS5 OE and KD). Finally, can you provide any data suggesting that GAS5 is indeed interacting with the IRF4 promoter region (as suggested in the model - Fig. S6)?

Given the constructive referee comments, we would like to invite you to revise your manuscript with the understanding that all referee/editor concerns must be fully addressed in the revised manuscript with additional data or new experiments and in a complete point-by-point response. Acceptance of your manuscript will depend on a positive outcome of a second round of review. It is EMBO reports policy to allow a single round of revision only and acceptance or rejection of the

manuscript will therefore depend on the completeness of your responses included in the next, final version of the manuscript.

REFEREE REPORTS

Referee #1:

Multiple sclerosis (MS) represents the multifocal and inflammatory disease in central nervous system (CNS). Microglia is involved in the inflammatory process of MS. Recent studies show that two types of macrophage/microglia are functionally classified, M1 and M2. In this study, authors identified GAS5, candidate non-coding gene, as an epigenetic regulator of microglia polarization, especially in M2 polarization.

Providing data in the manuscript are convinced and experiments are well designed.

Major concern in this study is the involvement of GAS5 in microglia. GAS5 is characterized and well-known lncRNA that promotes apoptosis and growth arrest. There are several isoforms and widely expressed in various tissues. As the authors examined, the interaction between GAS5 and glucocorticoid receptor (GR) was not observed in their system. Previous study by Kino et al reported that the 3' side region of GAS5 interacts to GR. This discrepancy for the different binding capacity of GAS5 toward GR should be explained in discussion section.

The expression of GAS5 is higher in the microglia of aged mice. What is the transcriptional regulator of GAS5?

In the experiment of the GAS5 over-expression, how much amount of GAS5 is expressed?

The knockdown efficiencies are almost half compared to those of control siRNA in Fig. S1A. Is it expected the similar effect if one allele of GAS5 genome is disrupted by CRISPER/Cas9 system?

Another lncRNA, HOTAIR, also interacts and recruits PRC2 complex through the terminal secondary structure of HOTAIR. Is it possible to compete or cooperate in microglia?

In Figure 5, the authors show the data of FISH analysis of GAS5 RNA. The most of signals are observed in cytoplasm. However, the proposed model by authors represents that GAS5 plays a role in nucleus (Fig. S6).

Abbreviation: check the CNS appearance.

Referee #2:

The authors used microarrays to identify 120 lncRNAs in mouse microglia that were polarised into M2-like cells by IL-4 in vitro in comparison to untreated microglia. Among these GAS5 as well as other known polarisation regulators were decreased. They confirmed in vitro that M2 polarisation of microglia seems to correlate with GAS5 downregulation. Furthermore, overexpression of GAS5 in microglia led to an inhibition of M2-like polarisation and promotion of M1-like polarisation while GAS5 inhibition had the opposite effect. These modulations had also respective functional effects on OPC and neurons in vitro by influencing apoptosis and differentiation. Analysis in EAE revealed a correlation of GAS5 expression with TNF α (M1) expression suggesting that GAS5 is also involved in microglia/macrophage polarisation in vivo. To further corroborate their data the authors transplanted GAS5 overexpressing/interfering microglia into the ventricles of EAE animals and observed an exacerbation/alleviation of the disease corresponding to increased/reduced inflammatory cell infiltration. GAS5 inhibited microglia also promoted remyelination in the lyssolecithin model. Finally the authors tried to identify the molecular pathway how GAS5 expression may influence microglia polarisation and identify IRF4 transcription and binding to PRC2 as potential mechanisms. In order to transfer these results to human disease GAS5 overexpression in human microglia is shown to have similar effects on polarisation in vitro and microglia/macrophages in MS lesions show a similar correlation between ramification and GAS5 expression as in the EAE experiments.

The manuscript is well written and the message is clear. The experiments are presented in a convincing manner. The topic of microglia polarisation during remyelination is timely and interesting. The authors present new findings and explore their observation in depth and thus reporting novel findings that are of great interest for scientists in the field of neurobiology/glia research/remyelination. There are only a few points that should be considered:

- In the transplantation experiments the modified microglia were injected into the ventricles and the analysis of the lesions was performed in the spinal cord. Do the transplanted microglia migrate from the ventricle to the spinal cord? It would be important if the transplanted cells can be detected in the lesions. Otherwise the effect is difficult to explain.

- The authors claim for their in vivo experiments (EAE, lysolecithin, MS lesions) that they investigated GAS5 expression in microglia. However, with their methods they can not differentiate between CNS intrinsic microglia and infiltrating macrophages. Thus they should rather claim that the results refer to microglia/macrophages. Or there may be a possibility to include recently published microglia specific marker like *Tmem119* or *Sall1*.

- In the results from the lysolecithin model experiments the authors state that LFB staining showed decreased myelin loss at 11 and 15 days post transplantation. At this stage there is no demyelination ongoing in this model but there is remyelination. Thus this should be increased remyelination rather than decreased demyelination.

- Sprague Dawley rats were used for OPC cell cultures while C57/BL6 mice were the source of microglia and neurons. Why were not also mouse OPC cultured? As there may be a species difference for many of the (investigated) factors this does not make sense.

- The comparison between microglia from young and old animals (Fig. 1 g, h, i) are interesting, but do not contribute to the overall findings described here. In addition, the authors do not follow up on this afterwards. Thus this reviewer thinks that these results should be omitted.

Referee #3:

The manuscript attributes a key role to GASS in inhibiting microglia polarization to the M2 phenotype.

While the overall message is interesting and important, this reviewer has major concerns regarding the presented data and the flow of the manuscript. Overall, each experiment seems to be well performed; however, the entire story is mainly built as a collection of circumstantial evidence regarding the relevance to EAE, rather than causal relationships.

Specifically:

Figure 1. These results are not unique to microglia. The authors should also culture monocytes. In addition, polarization towards M1 using IFN- γ is needed for comparison

The figures dealing with conditioned media are descriptive and add very little to the story, since conditioned media of M1 and M2 would have the same effect.

Fig 5. There is no way to know whether the Iba1+ cells are microglia or macrophages.

Figure 6. It is not clear whether there is a significant difference in disease severity.

Figure 6 and Figure 7. A more appropriate experiment would be to silence GASS in the resident microglia, rather than injecting exogenous microglia. Also, it is important to identify the homing sites of the injected microglia.

Thank you very much for your letter and the comments from the referees on our manuscript submitted to EMBO Reports [Paper # EMBOR-2016-43668V1]. We appreciated the reviewers' comments that "Providing data in the manuscript are convinced and experiments are well designed", "The manuscript attributes a key role to GAS5 in inhibiting microglia polarization to the M2 phenotype" and "The authors present new findings and explore their observation in depth and thus reporting novel findings that are of great interest for scientists in the field of neurobiology/glia research/remyelination." We are thankful for the kind acknowledgements of our work.

Importantly, we have made corresponding changes to our manuscript point by point after carefully studying the reviewers' and your comments. Here, we submit the revised manuscript and a list of changes as follows. We marked all the changes in red in the revised manuscript.

The following is a point-to-point response to your and the reviewers' comments.

Response to the editor: We think that the data on the EZH2-GAS5 interaction and IRF4 regulation are rather preliminary and need further strengthening. Other PRC1/2 proteins should be tested as in Fig. 9a and 9b to show a specific interaction with EZH2. As it is claimed that recruitment of PRC2 complexes by GAS5 to IRF4 leads to repression, H3K27me3 and H3K4me3 patterns need to be analyzed by Chip in Fig. 9a and correlated with the activity state of IRF4 (under normal conditions and upon GAS5 OE and KD). Finally, can you provide any data suggesting that GAS5 is indeed interacting with the IRF4 promoter region (as suggested in the model - Fig. S6)?

Answer: Thank you for the questions. We will answer your question in three parts.

First, to further demonstrate the combination of GAS5 and PRC2, we used RbAp48, the main binding subunit of PRC2, to perform RIP experiments. The results showed a similar binding ratio about 30% as EZH2 which is the catalytic subunit of PRC2 (Fig. EV4c). While PRC2 catalyses the methylation of histone H3 at lysine 27, PRC1 catalyses the monoubiquitylation of histone H2A via the ubiquitin ligases RING1A and RING1B. Besides, we also performed RIP experiments with the anti-RING1A/B antibody (Abcam, ab32644), and the results showed that there was no specific binding between RING1A/B and GAS5.

Figure EV4. Analysis of the mechanism regulating microglial polarization by GAS5.

(c) RNA IP analysis between RbAp48 and GAS5. N = 3 experiments. *** $P < 0.001$ (Student's t -test). Data are shown as the mean \pm SD.

RIP with RING1A/B:

RNA IP analysis between RING1A/B and GAS5. N = 3 experiments. Student's *t*-test. Data are shown as the mean \pm SD.

We added these words in “**Results:**” Page 11 line 229: “RIP experiments with another antibody against the main binding subunit of PRC2, RbAp48, showed a similar phenomenon (Fig. EV4c)”

Second, the main function of PRC2 is to catalyze H3K27me₃, while the trimethylation of H3K4 is catalyzed by set/MLL family (*Int J Hematol. 2017;105(1):7-16*). The trimethylation of H3K27 often activates transcription while H3K4me₃ inhibits. We conducted ChIP experiments using anti-H3K27me₃ antibody (millipore, 17-622) and found H3K27me₃ was significantly enhanced at the IRF4 promoter in the microglia overexpressing GAS5 while decreased in the microglia knocking down GAS5, which is consistent with the binding between EZH2 and the IRF4 promoter (**Fig. 7f, g**). Besides, we also performed ChIP assays using anti-H3K4me₃ antibody (millipore, 17-614). The H3K4me₃ was decreased at the IRF4 promoter in the microglia overexpressing GAS5, but changed a little in the microglia knocking down GAS5.

Figure 7: GAS5 suppresses IRF4 transcription by binding PRC2 and inhibiting microglial M2 polarization.

(f, g) ChIP analysis of microglia transduced with the CtrlOE/GAS5OE (f) or Ctrli/GAS5i (g) lentivirus on the IRF4 promoter regions using the anti-H3K27me3 antibody, n = 3 experiments. ** $p < 0.01$, *** $P < 0.001$ versus control (Student's *t*-test). Data are shown as the mean \pm SD.

ChIP with H3K4me3 antibody:

ChIP analysis of microglia transduced with the CtrlOE/GAS5OE (a) or Ctrli/GAS5i (b) lentivirus on the IRF4 promoter regions using the anti-H3K4me3 antibody, n = 3 experiments. ** $p < 0.01$, *** $P < 0.001$ versus control (Student's *t*-test). Data are shown as the mean \pm SD.

We added these words in “**Results:**” Page 12 line 239: “ChIP using H3K27me3 antibody showed the methylation of H3K27 at the IRF4 promoter was changed accordingly (Fig. 7f, g).”

Finally, to prove the binding of GAS5 RNA to IRF4 promoter, we performed Chromatin Isolation by RNA Purification (ChIRP) (*J Vis Exp.* 2012, 61:e3912) using an anti-GAS5 RNA probe, and found that there was indeed a binding between GAS5 mRNA and IRF4 promoter region (Fig. 7e).

In Fig. 7e (see above): GAS5 suppresses IRF4 transcription by binding PRC2 and inhibiting microglial M2 polarization.

(e) ChIRP analysis of the binding between the IRF4 promoter and GAS5, n = 3 experiments. IGF1 served as a negative control. * $P < 0.05$, *** $P < 0.001$ versus control (Student's *t*-test). Data are shown as the mean \pm SD.

We added these words in “**Results:**” Pg 11 line 238: “ChIRP analysis further confirmed the binding between GAS5 and the promoter regions of IRF4 (Fig. 7e), and ChIP using H3K27me3 antibody showed the methylation of H3K27 at the IRF4 promoter was changed accordingly (Fig. 7f, g).”

We added these sentences in “**Materials and methods:**” Page 25 line 510: “**ChIP and Chromatin Isolation by RNA Purification (ChIRP).** ChIP was performed using an EZ ChIP™ Chromatin Immunoprecipitation Kit (17-371, Millipore) following the manufacturer's protocol. Quantitative PCR was conducted to detect DNA fragments binding with EZH2 or H3K27me3 (17-622, Millipore). Primers were designed to detect a ~2000 bp area before the transcriptional start site. The ratio of DNA binding with EZH2 versus total DNA was calculated and compared between groups.

ChIRP was performed according to a previous report [44]. Briefly, biotin-labeled GAS5 cRNA probes as used in FISH were transcribed with the Biotin RNA Labeling Mix (Roche) and T7 RNA polymerase (Thermo) *in vitro*, treated with RNase-free DNase I (Thermo), and purified with the RNA Purification Kit (Transgen). Primary cultured microglia were collected and sonicated as in ChIP with adding RNA inhibitors. Then the hybridization was performed with biotinylated cRNA probes at 60°C overnight in hybridization solution. After adding streptavidin magnetic beads (Roche) and incubating for 30 minutes at 37°C, beads were washed and bound chromatin were

isolated as in ChIP. Quantitative PCR was conducted to detect DNA fragments binding with GAS5 and the ratio of DNA binding with GAS5 versus total DNA was calculated.”

Reviewer #1:

Multiple sclerosis (MS) represents the multifocal and inflammatory disease in central nervous system (CNS). Microglia is involved in the inflammatory process of MS. Recent studies show that two types of macrophage/microglia are functionally classified, M1 and M2. In this study, authors identified GAS5, candidate non-coding gene, as an epigenetic regulator of microglia polarization, especially in M2 polarization.

Providing data in the manuscript are convinced and experiments are well designed.

Major concern in this study is the involvement of GAS5 in microglia. GAS5 is characterized and well-known lncRNA that promotes apoptosis and growth arrest. There are several isoforms and widely expressed in various tissues. As the authors examined, the interaction between GAS5 and glucocorticoid receptor (GR) was not observed in their system. Previous study by Kino et al reported that the 3' side region of GAS5 interacts to GR. This discrepancy for the different binding capacity of GAS5 toward GR should be explained in discussion section.

Answer: Thank you for the question. Your suggestion of an explanation in Discussion about the binding between GAS5 and GR is really needed. As reported in the work by Kino et al (*Sci Signal. 2010;3(107):ra8*), “*Gas5 interacts with ligand-activated GR through its DBD in the cytoplasm and comigrates with GR into the nucleus*”. Notably, as shown in Figure 2A in Kino’s work, no specific binding exists without adding Dexamethasone, the ligand of GR. Even at 10^{-11} M, the concentration equivalent to physiologic level of circulating cortisol, only a very slight higher binding of GR with GAS5 than control antibody was observed. Our results were consistent with Kino’s report, and only a very small portion (about 2.5%) of GAS5 was observed binding with GR in the general culture condition (10% FBS in DF12). Besides, serum free medium was used when cytokines like IL-4 or letivirus were added. So we think GR was not a significant participant in the regulation of microglial polarization by GAS5 without adding Dexamethasone.

Figure 2A in Kino’s work (*Sci Signal. 2010;3(107):ra8*)

We add the following words in “**Discussion:**” Page 14 line 281: “GAS5 has been shown to act as a “decoy” to combine with the DNA-binding domain of glucocorticoid receptor and block its interactions with downstream targets [28]. However, GAS only interacts with ligand-activated GR and only a very small portion of GAS5 binding with GR at physiologic level of circulating cortisol, while no binding exists without GR ligand. GAS5 was found regulating microglial polarization under serum free conditions, which indicates the role of GR in the regulation of microglial polarization by GAS5 is ignorable.”

The expression of GAS5 is higher in the microglia of aged mice. What is the transcriptional regulator of GAS5?

Answer: Thank you for the question. GAS5 belongs to the 5'-TOP family, and growth arrest or treatment with inhibitors of protein translation lead to inhibition of their degradation, resulting in accumulation of Gas5 RNA (*Mol Cell Biol* 1998;18:6897–6909). Transcriptionally, GAS5 is regulated by mTOR and PI3K pathway as we discussed in “Discussion”. However, it is mentionable that accumulation of gas5 mRNA also comes from inhibition of degradation by forming stable submonosomal particles with ribosome at the posttranscriptional level in growth arrested cells (*Mol Cell Biol*. 1992; 12(8): 3514-21; *Mol Cell Biol* 1998;18:6897–6909). The gene expression signature of aged mouse microglia showed molecules associated with “protein processing in endoplasmic reticulum” were down-regulated while ribosome mRNAs were up-regulated (*Acta Neuropathol Commun*. 2015;3 31). Since the expression of molecules involved in mTOR and PI3K pathway are not changed in aged mice, we think the relative low translation occupancy or high ribosome quantity contribute to the forming of more stable submonosomal particles and accumulation of gas5 mRNA. However, the detailed mechanism needs further exploration. Besides, as these results contribute little to this whole work, we have decided to remove this part from the main text according to Reviewer 2’ suggestion.

In the experiment of the GAS5 over-expression, how much amount of GAS5 is expressed?

Answer: Thank you for the question. According to the results detected by qPCR, the level of GAS5 mRNA increased more than 50 times in microglia added GAS5^{OE} lentiviral at a MOI = 10 versus control.

The knockdown efficiencies are almost half compared to those of control siRNA in Fig. S1A. Is it expected the similar effect if one allele of GAS5 genome is disrupted by CRISPR/Cas9 system?

Answer: This is an interesting question. It is quite hard to compare the effects of knocking down by shRNA with knocking out one allele of a gene, as the heterozygous one often fails to decrease the expression by half because of the compensatory overexpression by the left allele (for example: *Nature*. 1998;393(6683):377-81). As for GAS5, the loci transcribes not only GAS5, but also a lot of functional snoRNAs (*Mol Cell Biol*. 1998;18(12):6897-909), which impedes us to knockout GAS5 only. More work need to be done to determine the functional motif of GAS5, which make it practical to knockout a functional exon.

Another lncRNA, HOTAIR, also interacts and recruits PRC2 complex through the terminal secondary structure of HOTAIR. Is it possible to compete or cooperate in microglia?

Answer: Thank you for the question. It is reported that more than 20% lncRNAs can bind with PRC2 (*Proc Natl Acad Sci U S A*. 2009;106(28):11667-72), but only 120 lncRNAs were differentially expressed in IL-4-stimulated M2 microglia versus resting microglia in our microarray analysis. As for HOTAIR, the microarray results showed that the expression doesn’t change after adding IL-4 versus control. Besides, we constructed an overexpression plasmid encoding human HOTAIR and transfected it into human microglia by electroporation. QPCR results showed the expression of M1 makers TNF- α and IL-1 β increased, but the expression of M2 markers Arg-1 and CD206 also increased, showing a complicated role of HOTAIR in the regulation of microglial polarization.

qPCR results:

Quantitative PCR analysis of M1 and M2 markers in human microglia nucleo-transfected with the human HOTAIR plasmid versus the control, n = 3 experiments. * $P < 0.05$, ** $p < 0.01$, *** $P < 0.001$. Student's t -test. Data are shown as the mean \pm SD.

In Figure 5, the authors show the data of FISH analysis of GAS5 RNA. The most of signals are observed in cytoplasm. However, the proposed model by authors represents that GAS5 plays a role in nucleus (Fig. S6).

Answer: Thank you for the question. In accordance with the previous report (*Sci Signal*. 2010;3(107):ra8), GAS5 was found expressed in both cytoplasm and nucleus in our work. We proved that GAS5 played a role in regulating the transcription of IRF4 by binding with PRC2 in the nucleus, but it is not the only function of GAS5. GAS5 has been reported binding with GR as an antagonist in both cytoplasm and nuclear and adsorbing miR-21 as a sponge (*Cell Death Differ*. 2013;20(11):1558-68). Failed to detect proteins that specifically bound to GAS5 by RNA pulldown, we were attracted by that a high proportion of GAS5 bound with EZH2. The role of GAS5 cooperated with PRC2 in nucleus explained its effects in microglial polarization well, but it can't be the whole story. The GAS5 located in the cytoplasm may also play a role in an unknown way, which needs to be further explored.

Abbreviation: check the CNS appearance.

Answer: Thank you for pointing out the mistake. We have corrected the abbreviation.

- Page 3 line 42: "Multiple sclerosis (MS) is a multifocal, inflammatory demyelinating disease of the CNS central nervous system (CNS)."
- Page 3 line 48: "As the major innate immune cells in the CNS central nervous system (CNS)"

Reviewer #2:

The authors used microarrays to identify 120 lncRNAs in mouse microglia that were polarised into M2-like cells by IL-4 in vitro in comparison to untreated microglia. Among these GAS5 as well as other known polarization regulators were decreased. They confirmed in vitro that M2 polarization of microglia seems to correlate with GAS5 downregulation. Furthermore, overexpression of GAS5 in microglia led to an inhibition of M2-like polarisation and promotion of M1-like polarisation while GAS5 inhibition had the opposite effect. These modulations had also respective functional effects on OPC and neurons in vitro by influencing apoptosis and differentiation. Analysis in EAE revealed a correlation of GAS5 expression with TNF α (M1) expression suggesting that GAS5 is also involved in microglia/macrophage polarisation in vivo. To further corroborate their data the authors transplanted GAS5 overexpressing/interfering microglia into the ventricles of EAE animals and observed an exacerbation/alleviation of the disease corresponding to increased/reduced inflammatory cell infiltration. GAS5 inhibited microglia also promoted remyelination in the lysolecithin model. Finally the authors tried to identify the molecular pathway how GAS5 expression may influence microglia polarisation and identify IRF4 transcription and binding to PRC2 as potential mechanisms. In order to transfer these results to human disease GAS5 overexpression in human microglia is shown to have similar effects on polarisation in vitro and microglia/macrophages in MS lesions show a similar correlation between ramification and GAS5 expression as in the EAE experiments.

The manuscript is well written and the message is clear. The experiments are presented in a convincing manner. The topic of microglia polarisation during remyelination is timely and interesting. The authors present new findings and explore their observation in depth and thus reporting novel findings that are of great interest for scientists in the field of neurobiology/glia research/remyelination. There are only a few points that should be considered:

- In the transplantation experiments the modified microglia were injected into the ventricles and the analysis of the lesions was performed in the spinal cord. Do the transplanted microglia migrate from the ventricle to the spinal cord? It would be important if the transplanted cells can be detected in the lesions. Otherwise the effect is difficult to explain.

Answer: Thank you for the question. The method that transplanting modified microglia to study the effects on EAE was used in many well-written research papers (*Immunity*. 2012;37(2):249-63; *J Clin Invest*. 2006;116(4):905-15). Besides, we performed anti-GFP immunohistochemistry 30 days postimmunization to confirm the localization of graft microglia in spinal cord in our previous work (*J Neurosci*. 2015;35(16):6350-65). Many GFP⁺ microglia were found in the spinal cord.

Representative anti-GFP IHF analysis of spinal cord sections from microglia-transplanted EAE mice at 30dpi. Scale bars = 50 μ m.

- The authors claim for their in vivo experiments (EAE, lysolecithin, MS lesions) that they investigated GAS5 expression in microglia. However, with their methods they cannot differentiate between CNS intrinsic microglia and infiltrating macrophages. Thus they should rather claim that the results refer to microglia/macrophages. Or there may be a possibility to include recently published microglia specific marker like Tmem119 or Sall1.

Answer: Thank you for the question. Iba1 is a classical marker to label microglia/macrophage, and microglia are often viewed as macrophages settled in the CNS. They are regulated in a similar way. Actually, we claimed the two cell types are indistinguishable by Iba1 as “Iba1⁺ cells (microglia and infiltrating macrophages)” in the text. Besides, we performed FISH staining with Tmem119 antibody according to your suggestion and found a similar expression pattern of GAS5 in mem119⁺ cells as GAS5 was highly expressed in amoeba-shaped microglia (Fig. EV3).

Following are the figures:

Figure EV3. GAS5 is expressed in the microglia of EAE.

Representative FISH analysis of GAS5 (green) co-stained with an anti-TMEM119 antibody (red) in spinal cord sections from EAE mice at 30dpi. Scale bars = 25 μ m.

We also added these words in “**Results:**” Page 8 line 162: “The same went for TMEM119⁺ microglia (Fig. EV3)”

- In the results from the lysolecithin model experiments the authors state that LFB staining showed decreased myelin loss at 11 and 15 days post transplantation. At this stage there is no demyelination ongoing in this model but there is remyelination. Thus this should be increased remyelination rather than decreased demyelination.

Answer: Thank you for the question. We corrected the expression to "increased remyelination".

Page 10 line 199: “LFB staining showed increased remyelination decreased myelin loss at 11 dpl and 15 dpl after MG^{GAS5i} transplantation (Fig. 6a, b).”

- Sprague Dawley rats were used for OPC cell cultures while C57/BL6 mice were the source of microglia and neurons. Why were mouse OPC not also cultured? As there may be a species difference for many of the (investigated) factors this does not make sense.

Answer: Thank you for the question. The cell culture technique of mouse OPCs *in vitro* is still immature now. The quantity of primary cultured mouse OPCs is quite small and the differentiation is inefficient. Because of these reasons, most present researches use the rat OPCs to replace (for example: *Nat Neurosci.* 2016;19(5):678-89; *Cell.* 2013;152(1-2):248-61). However, it is important to keep in mind that there may be a species difference between rat and mouse OPCs. As for this work, we used the LPC model to study the remyelination process in mouse, and the results were in accordance with the *in vitro* co-culture experiments using rat OPCs. We think the usage of rat OPCs does not influence the conclusion in this work.

- The comparison between microglia from young and old animals (Fig. 1 g, h, i) are interesting, but do not contribute to the overall findings described here. In addition, the authors do not follow up on this afterwards. Thus this reviewer thinks that these results should be omitted.

Answer: Thank you for the question. The relation between GAS5 expression level and growth state and the report of microglial M2 polarization disorder in aged mice provoked us to compare GAS5 expression between young and aged mouse microglia. The result indicated that the increased

expression of GAS5 may be one of the reasons. These results contribute little to this work actually, and, we have removed this part from the main text according to your suggestion.

Reviewer #3:

The manuscript attributes a key role to GASS in inhibiting microglia polarization to the M2 phenotype. While the overall message is interesting and important, this reviewer has major concerns regarding the presented data and the flow of the manuscript. Overall, each experiment seems to be well performed; however, the entire story is mainly built as a collection of circumstantial evidence regarding the relevance to EAE, rather than causal relationships. Specifically:

Figure 1. These results are not unique to microglia. The authors should also culture monocytes. In addition, polarization towards M1 using IFN- γ is needed for comparison

Answer: Thank you for the question. We transfected primary cultured monocytes with GAS5 over-expression or interference lentiviral vectors according to your suggestion, and the qPCR showed a similar effect as in microglial polarization. GAS5 over-expression in monocytes decreased the expression of the M2 makers Ym-1, Fizz-1, CD206 and IGF-1 and increased the expression of the M1 markers TNF- α and IL-1b. Conversely, interference with GAS5 in monocytes resulted in an increase in the expression of M2 markers and a decrease in the expression of M1 markers (Fig. EV5c, d). Besides, we detected the expression of GAS5 in microglia after adding IFN- γ for 24 hours, and the qPCR showed no significant change of the expression of GAS5, showing the IFN- γ /JAK/STAT pathway was not involved in the regulation of GAS5 expression. The following are the results.

Fig EV5c-d:

Figure EV5. GAS5 is highly expressed in aged microglia and regulates the inflammatory response of monocytes.

(c, d) Quantitative PCR analysis of M1 and M2 markers in monocytes transfected with the GAS5OE (c) or GAS5i (d) lentivirus vectors versus the control, $n \geq 3$ experiments. * $P < 0.05$, ** $p < 0.01$, *** $P < 0.001$ (c, d Student's t -test). Data are shown as the mean \pm SD.

We also added these words in “**Results:**” Page 6 line 125: “Over-expression or interference with GAS5 in peripheral monocytes resulted in a similar phenotype (Fig. EV5c, d).”

The figures dealing with conditioned media are descriptive and add very little to the story, since conditioned media of M1 and M2 would have the same effect.

Answer: Thank you for the question. To show the effects of GAS5-modified microglia to OPCs and neurons clearly in a simple system, we conducted the co-culture experiments *in vitro*. Generally, the conditioned medium from M1 polarized microglia is more toxic and aggravates injury, while medium from M2 polarized microglia is more cytoprotective and promotes repair (reports like *Nat Neurosci.* 2013 Sep;16(9):1211-8; reviewed by us in *Neurosci Bull* 2013, 29(2): 189–198). However, as you pointed out, polarization states cannot define the function of microglia, as functions like migration, phagocytosis and proliferation all affect the influence of microglia to other cells. The co-culture experiments are descriptive and auxiliary. We have removed this part away from the main text according to your suggestion.

Fig 5. There is no way to know whether the Iba1+ cells are microglia or macrophages.

Answer: Thank you for the question. As we have answered above, Iba1 is a classical marker to label microglia/macrophage, and microglia are often viewed as macrophages settled in the CNS. We claimed the two cell types are indistinguishable by Iba1 as “*Iba1⁺ cells (microglia and infiltrating macrophages)*” in the text. Generally, there is no need to distinguish them as they are regulated in a similar way. Besides, we performed FISH staining with Tmem119 antibody which has been reported specifically labelling microglia and found a similar expression pattern as GAS5 was highly expressed in amoeba-shaped Tmem119⁺ microglia (**Fig. EV3**). Following are the figures.

Fig EV3 [see above]: GAS5 is expressed in the microglia of EAE.

Representative FISH analysis of GAS5 (green) co-stained with an anti-TMEM119 antibody (red) in spinal cord sections from EAE mice at 30dpi. Scale bars = 25 μ m.

We also added these words in “**Results:**” Page 8 line 162: “The same went for TMEM119⁺ microglia (Fig. EV3)”

Figure 6. It is not clear whether there is a significant difference in disease severity.

Answer: Thank you for the question. According to the statistical analysis of EAE scores, there was a significant difference between different groups, although the difference is not too great.

Figures 6 and 7: A more appropriate experiment would be to silence GASS in the resident microglia, rather than injecting exogenous microglia. Also, it is important to identify the homing sites of the injected microglia.

Answer: Thank you for the question. We attempted to construct GAS5 knockout mice, but the GAS5 loci transcribes not only GAS5, but also a lot of functional snoRNAs (*Mol Cell Biol.* 1998;18(12):6897-909), which impedes us to knockout GAS5 only. We chose transplanting modified microglia to study the effects on EAE because the method has been proved effective and was used in many well-written research papers (*Immunity.* 2012;37(2):249-63; *J Clin Invest.* 2006;116(4):905-15). Besides, we performed anti-GFP immunohistochemistry to locate grafted microglia in spinal cord at 30 dpi in our previous work (*J Neurosci.* 2015;35(16):6350-65). Many GFP⁺ microglia were found in the spinal cord.

[See above for representative anti-GFP IHF analysis of spinal cord sections from microglia-transplanted EAE mice at 30dpi. Scale bars = 50 μ m.]

Thank you for the submission of your revised manuscript to our editorial offices. We have now received the report from the two referees that were asked to re-evaluate your study (you will find enclosed below). Referee #2 was not able to look at the manuscript again. As you will see, both referees now support the publication of your manuscript in EMBO reports. I also acknowledge that you have sufficiently addressed my specific requests.

Before we can proceed with formal acceptance, the following editorial requests need to be addressed in a final revised version of the manuscript.

The title of the paper currently contains too many abbreviations and is too long. Please provide a simpler and shorter title (not more than 100 characters including spaces).

As all the Western blot panels show significantly cropped images (Figures: 2E, 2F, 7B, 7K, 7L, 7M, EV2C and EV2D), we would require the original source data for these published together with the paper (with the aim of making primary data more accessible and transparent to the reader). The source data will be published in a separate source data files online along with the accepted manuscript and will be linked to the relevant figure. Please submit the source data (scans of the entire gels or blots) of your key experiments together with the revised manuscript. Please include size markers for scans of entire gels, label the scans with figure and panel number, and send one PDF file per figure or per figure panel.

Please also provide higher resolution images for the Western blot images (Figures: 2E, 2F, 7B, 7K, 7L, 7M, EV2C and EV2D).

You provided additional data responding to my comments. Please add these data to the manuscript and discuss them, either as main figures or as EV figure (or as part of the Appendix). I refer to: "RIP with RING1A/B" and "ChIP with H3K4me3 antibody".

Please try to reduce the number of main figures to 8 (either by fusing figure panels, or by moving panels to EV figures).

Please rename the items in the appendix. There should be figures and tables that are independently numbered (Figure S1, S2, S3 etc. and Table S1, S2, S3). Please include page numbers and put them in the Table of Contents. After renaming, please change the callouts accordingly in your manuscript text.

Where the results of the microarray screen deposited and are available? Please indicate database and accession number.

Please move the keywords to page 1, below the running title.

Finally, please have the manuscript edited by a native English speaker.

Please note that we now mandate that all corresponding authors list an ORCID digital identifier! We would therefore need the ORCID digital identifier for Li Cao & Cheng He.

I look forward to seeing a revised version of your manuscript when it is ready. Please let me know if you have questions or comments regarding the revision.

REFEREE REPORTS

Referee #1:

The authors have addressed all concerns in the revised version in a satisfactory manner. Thus, I think that this manuscript is suitable for publication in EMBO reports.

Referee #2:

The authors have adequately responded to the reviewers comments.

Thank you very much for your letter and the comments on our revised manuscript submitted to EMBO Reports. It really excited us that “both referees now support the publication of your manuscript in EMBO reports. I also acknowledge that you have sufficiently addressed my specific requests.” We are thankful for the kind acknowledgements of our work.

Importantly, we have made corresponding changes to our manuscript point by point after carefully studying your comments. Here, we submit the revised manuscript and a list of changes as follows. We mark all the changes in red in the revised manuscript.

Thank you very much for the excellent and professional revision of our manuscript.

The following is a point-to-point response to your comments:

The title of the paper currently contains too many abbreviations and is too long. Please provide a simpler and shorter title (not more than 100 characters including spaces).

Answer: Thank you for the question. We have shortened the title as “Long Noncoding RNA GAS5 Inhibits Microglial M2 Polarization and Exacerbates EAE” (79 characters including spaces).

As all the Western blot panels show significantly cropped images (Figures: 2E, 2F, 7B, 7K, 7L, 7M, EV2C and EV2D), we would require the original source data for these published together with the paper

Answer: Thank you for the questions. We have uploaded the original source data for all the above images.

Please also provide higher resolution images for the Western blot images (Figures: 2E, 2F, 7B, 7K, 7L, 7M, EV2C and EV2D).

Answer: Thank you for the question. We noticed that the figure guidelines on your website said allow for printing at a resolution of at least 300 pixel per inch (ppi), and our figure were made at 600ppi. If higher resolution images were needed, we are happy to either change the figure as your clear requirements or offer the original data file.

You provided additional data responding to my comments. Please add these data to the manuscript and discuss them, either as main figures or as EV figure (or as part of the Appendix). I refer to: "RIP with RING1A/B" and "ChIP with H3K4me3 antibody."

Answer: Thank you for the questions. We have added these data to the manuscript and discussed them. "RIP with RING1A/B" was added as “Figure EV4d” and "ChIP with H3K4me3 antibody" was added as “Appendix figure S2.”

We add the following words in Results:

Page 11 line 200: In contrast, RIP experiments with the anti-RING1A/B (subunits of PRC1) antibody showed no specific binding existed (Fig. EV4d)

Page 12 line 217: Besides, we analyzed another epigenetic marker by performing ChIP assays using anti-H3K4me3 antibody and found the H3K4me3 was decreased at the IRF4 promoter in MGGAS5OE, which was in accordance with the inhibited transcription, but changed a little in MGGAS5i (Appendix figure S2)

We added the following words in the Discussion:

Page 15 line 270: Specifically, PRC1, another polycomb repressive complex, showed no specific binding with GAS5, supporting the more important role of PRC2 in recruiting lncRNAs.

Page 15 line 277: Besides, another epigenetic marker, H3K4me3, also changed in MGGAS5OE at the IRF4 promoter, but changed a little in MGGAS5i, indicating more regulating mechanisms need to be explored

Please try to reduce the number of main figures to 8

Answer: Thank you for the question. We have removed ifigure 9i and changed it as the schematic summary figure.

Please rename the items in the appendix. There should be figures and tables that are independently numbered (Figure S1, S2, S3 etc. and Table S1, S2, S3). Please include page numbers and put them in the Table of Contents. After renaming, please change the callouts accordingly in your manuscript text.

Answer: Thank you for the questions. We have changed all of them according to your suggestions.

Where the results of the microarray screen deposited and are available? Please indicate database and accession number.

Answer: Thank you for the question. We have deposited the data to GEO and the accession number is GSE99216.

Please move the keywords to page 1, below the running title.

Answer: Thank you for the question. We have changed this according to your suggestion.

Finally, please have the manuscript edited by a native English speaker.

Answer: Thank you for the question. As non-native speakers of English, we paid great attention to the writing. Before submitting to the journal, we have polished the manuscript by the qualified company "American Journal Experts" as well as our collaborator Robin J. M. Franklin who is a native English and a professor working in University of Cambridge. If our manuscript needs more modification, please point out the problems directly. Your kind help will be highly appreciated.

Referee #1:

The authors have addressed all concerns in the revised version in a satisfactory manner. Thus, I think that this manuscript is suitable for publication in EMBO reports.

We thank the kind acknowledgement of our work.

Referee #2:

The authors have adequately responded to the reviewers comments.

We thank the kind acknowledgement of our work.

3rd Editorial Decision

09 June 2017

Thank you for the submission of your revised manuscript to our editorial offices. Before we can proceed with formal acceptance, these further editorial requests need to be addressed:

Thank you for shortening the title. However, the title "Long Noncoding RNA GAS5 Inhibits Microglial M2 Polarization and Exacerbates EAE" still contains too many abbreviations. In particular, many readers will not know what EAE stands for. Maybe, one could call this animal model of multiple sclerosis. Please try to come up with a new title without EAE.

Please upload the abstract written in present tense.

I think the Western blot panels and their source data are fine, except for Fig. 7B. This looks very grainy and out of focus. Can you provide a better image for this blot and the respective source data (maybe from a replicate experiment)?

Thank you for adding the additional data to the manuscript. However, I would suggest to slightly changing the new text you added:

Page 11 line 200: "In contrast, RIP experiments with an anti-RING1A/B (subunits of PRC1) antibody showed no specific interaction (Fig. EV4d)".

Page 12 line 217: "In addition, we analysed another epigenetic histone mark by performing ChIP assays using an anti-H3K4me3 antibody. The results show that H3K4me3 was decreased at the IRF4 promoter in MGGAS5OE, but did not change in MGGAS5i (Appendix Figure S2), which is in accordance with the inhibited transcription observed upon GAS5 overexpression."

Page 15 line 270: "Importantly, components of the Polycomb repressive complex 1 (PRC1), showed no interaction with GAS5, supporting a specific role for PRC2 in recruiting lncRNAs."

Page 15 line 277: "Further, the active epigenetic mark H3K4me3, was reduced in MGGAS5OE at the IRF4 promoter, but showed little changes in MGGAS5i, confirming the reduced expression of IRF4 observed upon GAS5 overexpression, and indicating that further regulative chromatin modifying mechanisms act during this process."

The appendix file is still a bit messy. Please call this file Appendix, and use only black fonts. For simplicity, please turn the item now called "Appendix 1" (the sequence of FISH probes) into a table (Appendix Table S1), and rename the other tables accordingly, and update the callouts in the manuscript text.

Please add a section to the Methods section of the main manuscript text called "Data Deposition" and mention there that the results of the microarray have been uploaded to GEO with the accession number GSE99216.

Finally, we need the ORCID IDs for Li Cao and Cheng He to be linked to their profiles on our website. This can only be done by the authors themselves. They need to log in and in their profile there should be a button to link the IDs. If you have problems regarding this (or any other questions), please contact our editorial assistant.

I look forward to seeing a revised version of your manuscript when it is ready. Please let me know if you have questions or comments regarding the revision.

3rd Revision - authors' response

16 July 2017

Thank you very much for your letter and the comments on our revised manuscript submitted to EMBO Reports [Paper # EMBOR-2016-43668V2]. We have made corresponding changes to our manuscript point by point following your suggestions. Here, we submit the revised manuscript and a list of changes as follows. We made all the changes under "revision mode" in the revised manuscript. Thank you very much for the excellent and professional revision of our manuscript.

The following is a point-to-point response to your comments:

Thank you for shortening the title. However, the title "Long Noncoding RNA GAS5 Inhibits Microglial M2 Polarization and Exacerbates EAE" still contains too many abbreviations. In particular, many readers will not know what EAE stands for. Maybe, one could call this animal model of multiple sclerosis. Please try to come up with a new title without EAE.

Answer: Thank you for the question. We have modified the title as "GAS5 Inhibits Microglial M2 Polarization and Exacerbates the Animal Model of Multiple Sclerosis" (95 characters including spaces).

Please upload the abstract written in present tense.

Answer: Thank you for the questions. We have revised the abstract.

I think the Western blot panels and their source data are fine, except for Fig. 7B. This looks very grainy and out of focus. Can you provide a better image for this blot and the respective source data (maybe from a replicate experiment)?

Answer: Thank you for the question. We have re-conducted the RNA pulldown experiment and gained the new figure. We replaced Fig. 7B and changed the original source data accordingly.

Thank you for adding the additional data to the manuscript. However, I would suggest to slightly changing the new text you added:

Page 11 line 200: "In contrast, RIP experiments with an anti-RING1A/B (subunits of PRC1) antibody showed no specific interaction (Fig EV4d)."

Page 12 line 217: "In addition, we analysed another epigenetic histone mark by performing ChIP assays using an anti-H3K4me3 antibody. The results show that H3K4me3 was decreased at the IRF4 promoter in MGGAS5OE, but did not change in MGGAS5i (Appendix Figure S2), which is in accordance with the inhibited transcription observed upon GAS5 overexpression."

Page 15 line 270: "Importantly, components of the Polycomb repressive complex 1 (PRC1), showed no interaction with GAS5, supporting a specific role for PRC2 in recruiting lncRNAs."

Page 15 line 277: "Further, the active epigenetic mark H3K4me3, was reduced in MGGAS5OE at the IRF4 promoter, but showed little changes in MGGAS5i, confirming the reduced expression of IRF4 observed upon GAS5 overexpression, and indicating that further regulative chromatin modifying mechanisms act during this process."

Answer: Thank you for the kind modification of our description. We have changed the manuscript according to your suggestions.

The appendix file is still a bit messy. Please call this file Appendix, and use only black fonts. For simplicity, please turn the item now called "Appendix 1" (the sequence of FISH probes) into a table (Appendix Table S1), and rename the other tables accordingly, and update the callouts in

Answer: Thank you for the question. We have changed all of them according to your suggestions.

Please add a section to the Methods section of the main manuscript text called "Data Deposition" and mention there that the results of the microarray have been uploaded to GEO with the accession number GSE99216.

Answer: Thank you for the suggestion. We have added this section in the revised manuscript. Page 27 line 511: **"Data Deposition** All microarray data have been deposited in the Gene Expression Omnibus (GEO GSE99216)."

Finally, we need the ORCIDs for Li Cao and Cheng He to be linked to their profiles on our website.

Answer: Thank you for the question. They have linked their ORCIDs with the profiles accordingly.

I am very pleased to accept your manuscript for publication in the next available issue of EMBO reports. Thank you for your contribution to our journal.

YOU MUST COMPLETE ALL CELLS WITH A PINK BACKGROUND

Corresponding Author Name: Li Cao & Cheng He

Manuscript Number: EMBOR-2016-43668V1